# Interfacial alloying between lead halide perovskite crystals and hybrid glasses

Xuemei Li[1], Wengang Huang[1], Andraž Krajnc [2], Yuwei Yang[3], Atul Shukla [4], Jaeho Lee [1], Mehri Ghasemi[5], Isaac Martens [6], Bun Chan [7], Dominique Appadoo[8], Peng Chen[9], Xiaoming Wen [5], Julian A. Steele[4,9], Haira G. Hackbarth[3], Qiang Sun[10,11], Gregor Mali [2], Rijia Lin[1], Nicholas M. Bedford[3], Vicki Chen[1,12], Anthony K. Cheetham [13], Luiz H. G. Tizei [14], Sean M. Collins [15], Lianzhou Wang [1,9] & Jingwei Hou [1] ✉

The stellar optoelectronic properties of metal halide perovskites provide enormous promise for next-generation optical devices with excellent conversion efficiencies and lower manufacturing costs. However, there is a longstanding ambiguity as to whether the perovskite surface/interface (*e.g.* structure, charge transfer or source of off-target recombination) or bulk properties are the more determining factor in device performance. Here we fabricate an array of $CsPbI_3$ crystal and hybrid glass composites by sintering and globally visualise the property-performance landscape. Our findings reveal that the interface is the primary determinant of the crystal phases, optoelectronic quality, and stability of $CsPbI_3$. In particular, the presence of a diffusion "alloying" layer is discovered to be critical for passivating surface traps, and beneficially altering the energy landscape of crystal phases. However, high-temperature sintering results in the promotion of a non-stoichiometric perovskite and excess traps at the interface, despite the short-range structure of halide is retained within the alloying layer. By shedding light on functional hetero-interfaces, our research offers the key factors for engineering high-performance perovskite devices.

Lead halide perovskites feature exceptional optoelectronic properties such as high light-to-energy conversion, emission colour purity, and photoluminescence quantum yield. These characteristics make them a popular candidate for next-generation semiconducting materials to be used within solar panels, sensors, and light-emitting diodes (LEDs)[1–3]. Great effort has been spent in uncovering the mechanistic aspects that contribute to their exceptional performance and finding the origin of their large absorption coefficients, low exciton binding energies, and

[1]School of Chemical Engineering, The University of Queensland, St Lucia, QLD 4072, Australia. [2]Department of Inorganic Chemistry and Technology, National Institute of Chemistry, 1001 Ljubljana, Slovenia. [3]School of Chemical Engineering, The University of New South Wales, Kensington, NSW 2052, Australia. [4]School of Mathematics and Physics, The University of Queensland, St Lucia, QLD 4072, Australia. [5]School of Science, RMIT University, Melbourne, VIC 3000, Australia. [6]European Synchrotron Radiation Facility, 71 Avenue des Martyrs, 38000 Grenoble, France. [7]Graduate School of Engineering, Nagasaki University, Nagasaki 852-8521, Japan. [8]Australian Synchrotron, 800 Blackburn Rd, Clayton, VIC 3168, Australia. [9]Australian Institute for Bioengineering and Nanotechnology, The University of Queensland, St Lucia, QLD 4072, Australia. [10]State Key Laboratory of Oral Diseases, National Clinical Research Center for Oral Diseases, West China Hospital of Stomatology, Sichuan University, Chengdu, Sichuan 610041, China. [11]Sichuan Provincial Engineering Research Center of Oral Biomaterials, Chengdu, Sichuan 610041, China. [12]University of Technology Sydney, 15 Broadway, Ultimo, NSW 2007, Australia. [13]Materials Research Laboratory, University of California, Santa Barbara, CA 93106, USA. [14]Université Paris-Saclay, CNRS, Laboratoire de Physique des Solides, 91405 Orsay, France. [15]School of Chemical and Process Engineering and School of Chemistry, University of Leeds, Leeds LS2 9JT, UK. ✉e-mail: jingwei.hou@uq.edu.au

long carrier diffusion lengths and lifetimes[4–6]. Although immense effort has been devoted to device development, a practical application of perovskites is still hampered by their low stability in multiple aspects[7,8]. For instance, in addition to being susceptible to light, polar solvents and oxygen, the hybrid perovskite formamidinium lead iodide (FAPbI$_3$), which is popular in solar energy conversion, is susceptible to cation sublimation[9,10]. Although inorganic perovskites like CsPbI$_3$ overcome thermal stability issues, polymorphism remains a key challenge, where the optoelectronically active α-, β- and γ-phases exist exclusively at elevated temperatures (>250 °C), with the inactive δ-phase being thermodynamically favoured at ambient temperatures[6,11].

One common strategy for addressing these challenges associated with pure perovskite compounds is to composite them with secondary components[12,13]. Within this context, organic ligands, polymers, inorganic zeolites and glasses, and recently metal–organic frameworks (MOFs) have all been studied, with each type having its own advantages, and simultaneously disadvantages[14–18]. Understanding the behaviour and function of the interface inside the composites is critical. Such knowledge will further aid in the development of more efficient and stable devices: perovskites are typically sandwiched between electron and hole transport layers in LEDs and solar panel systems, and the properties of these interfaces can be critical to device performance[19]. Surface traps have long been a source of confusion and usually detrimental to the performance and stability of perovskites[20], even though they are typically considered as defect-tolerant materials[21]. On the other hand, chemical disorder can also capture diffusing carriers over micrometre-length scales, resulting in radiative recombination, outcompeting the capture of carriers in more electronically disordered and trap-rich regions[22]. However, studying both positive and negative effects from the interface remains a difficult task since most high-performing composites are not sufficiently stable to keep their original properties/functions against prolonged exposure to handling and inspection[23,24]. Other composites may lack a distinct chemical and/or physical contrast between two phases, reducing the capacity to extract important interfacial information[25].

MOF hybrid glass perovskite composites offer a unique opportunity to pinpoint the interface. The rich structural, chemical, and physical features, as well as the contrasts across phases, allow for the controlled isolation and investigation of the interfaces in composites and the establishment of property-performance correlation. In particular, these high performing composites are stable for characterisation by electron, synchrotron X-ray, and Terahertz beams and storage under ambient conditions[26]. Herein, we fabricated a series of hybrid glassy perovskite composites via sintering, and uncovered the origin of their exceptional performance. The atomic-level chemical and structural profile of the CsPbI$_3$ and hybrid glass composite are mapped using a combination of advanced characterisations, revealing the controllable formation of interdiffusion alloying layer during sintering. The layer stabilises CsPbI$_3$ optoelectronic phases and passivates its trap states. On the other hand, alloying can create non-stoichiometric perovskite regions and quenches photoluminescence after high-temperature sintering, despite the preservation of short-range rigid halide structures in the nanometre-thick alloying layer (Fig. 1a).

## Results

### Fabrication of composites

The crystal-glass composites were fabricated with pre-formed a$_g$ZIF-62 (a$_g$ refers to amorphous glass) [Zn(Im)$_{1.95}$(bIm)$_{0.05}$] (Im: imidazolate and bIm: benzimidazolate) and mechanochemically produced CsPbI$_3$ powders, with detailed procedures in Supplementary Information. Given the polymorphism nature of CsPbI$_3$, computational chemistry (at the xTB level) was applied to investigate its energy profiles. First, we calculated the energies of pure α (cubic) and δ (orthorhombic) CsPbI$_3$ with periodic boundary conditions, finding that the δ-phase was more stable by 18 kJ mol$^{-1}$ per CsPbI$_3$ unit. This is qualitatively consistent with

experimental observations and provides a degree of confidence for the use of xTB for further investigations. We then used several molecular cluster models to evaluate the energetics of surface formation to reveal the energy landscape of the CsPbI$_3$ within composites. Figure 1b shows two of the representative products between a fragment of the CsPbI$_3$ surface and an a$_g$ZIF-62 fragment. They were two of the most stable structures that we identified, with ligand exchange at the interface. The generation of the interfacial α-phase product was more exothermic than that of the δ-phase product. At the xTB level, the reaction energies were −1224 (α) and −1033 (δ) kJ mol$^{-1}$. The difference was sufficient to overcome the inherent stability of the δ-phase over the α-phase (18 kJ mol$^{-1}$ per CsPbI$_3$ unit, and thus by -150 kJ mol$^{-1}$ for the surface models with 8 or 9 units). The calculation results indicate the interfacial bond favours the formation of optoelectronically active phases of CsPbI$_3$ within composite.

An array of composites was then fabricated with different a$_g$ZIF-62 and CsPbI$_3$ ratios and sintering temperatures. ZIF-62 glass, instead of ZIF-62 crystal, was implemented to promote the formation of adaptive interfaces within the composites, given its higher enthalpy and lower temperature required to access the viscous flowing state[27]. The composite samples were hereby referred to as (CsPbI$_3$)(a$_g$ZIF-62) (X/Y) (prior to sintering) and (CsPbI$_3$)$_X$(a$_g$ZIF-62)$_Y$ (after sintering), where X and Y are the percentage mass of each component, with X values ranging from 1 to 85. Sintering pure CsPbI$_3$ at temperatures greater than 320 °C transformed the non-perovskite δ-phase into an optoelectronically active cubic perovskite α-phase, which returned to an inactive δ-phase during returning to ambient (Supplementary Fig. 1)[6,28]. Ex situ X-ray diffraction (XRD) revealed that the CsPbI$_3$ structures within the composites were highly dependent on its fraction and the sintering temperature (Fig. 1c and Supplementary Fig. 2). Prior to sintering, the powder mixtures exhibited poor crystallinity as expected for the mechanochemically synthesised CsPbI$_3$. Following the sintering process, CsPbI$_3$ crystallinity enhanced with several phases emerged on top of two a$_g$ZIF-62 broad peaks (centred near 15 and 35°). Higher sintering temperatures and lower CsPbI$_3$ loading, in general, can better preserve the CsPbI$_3$ perovskite phase. Figure 1d and Supplementary Fig. 3 shows a summary of the relative proportion of active γ-CsPbI$_3$ phase within the composites. After sintering at 350 °C, (CsPbI$_3$)$_{0.4}$(a$_g$ZIF-62)$_{0.6}$ can achieve nearly complete preservation of the γ-phase, and further increase of the CsPbI$_3$ loading resulted in a mixed active γ- and inactive δ-phases. In situ XRD was used to track the sintering-induced phase changes in (CsPbI$_3$)(a$_g$ZIF-62)(40/60) (Supplementary Fig. 4), showing the emergence of active phase CsPbI$_3$ from 150 °C, which was substantially lower than the intrinsic phase transition temperature of pure CsPbI$_3$ near equilibrium (320 °C). Further heating improved the crystallinity and the proportion of perovskite crystal formed. High resolution in situ synchrotron powder XRD further proved the phase evolutions are highly dependent on the loading of the perovskites, where the optoelectronically active phase can not be effectively preserved within composite when the CsPbI$_3$ loading is higher than 40% (Supplementary Fig. 5). After sintering, the composite showed a smooth and continuous surface morphology, with clear phase contrast under scanning electron microscope (SEM, Supplementary Fig. 6).

### Stable optical performance

The composites exhibited characteristic red-emitting PL under UV excitation, originated from the stable formations of active γ-CsPbI$_3$ phase (Supplementary Fig. 7), with their relative intensity of emissions charted in Supplementary Fig. 8. The overall PL maximum was realised by (CsPbI$_3$)$_{0.25}$(a$_g$ZIF-62)$_{0.75}$ at 275 °C sintering. The corresponding maximum intensity of the PL emission from composites with different CsPbI$_3$ loadings occurred at different sintering temperatures: generally, samples with greater CsPbI$_3$ loadings achieved their PL maxima at a lower temperature: e.g. (CsPbI$_3$)$_{0.01}$(a$_g$ZIF-62)$_{0.99}$ showed a PL

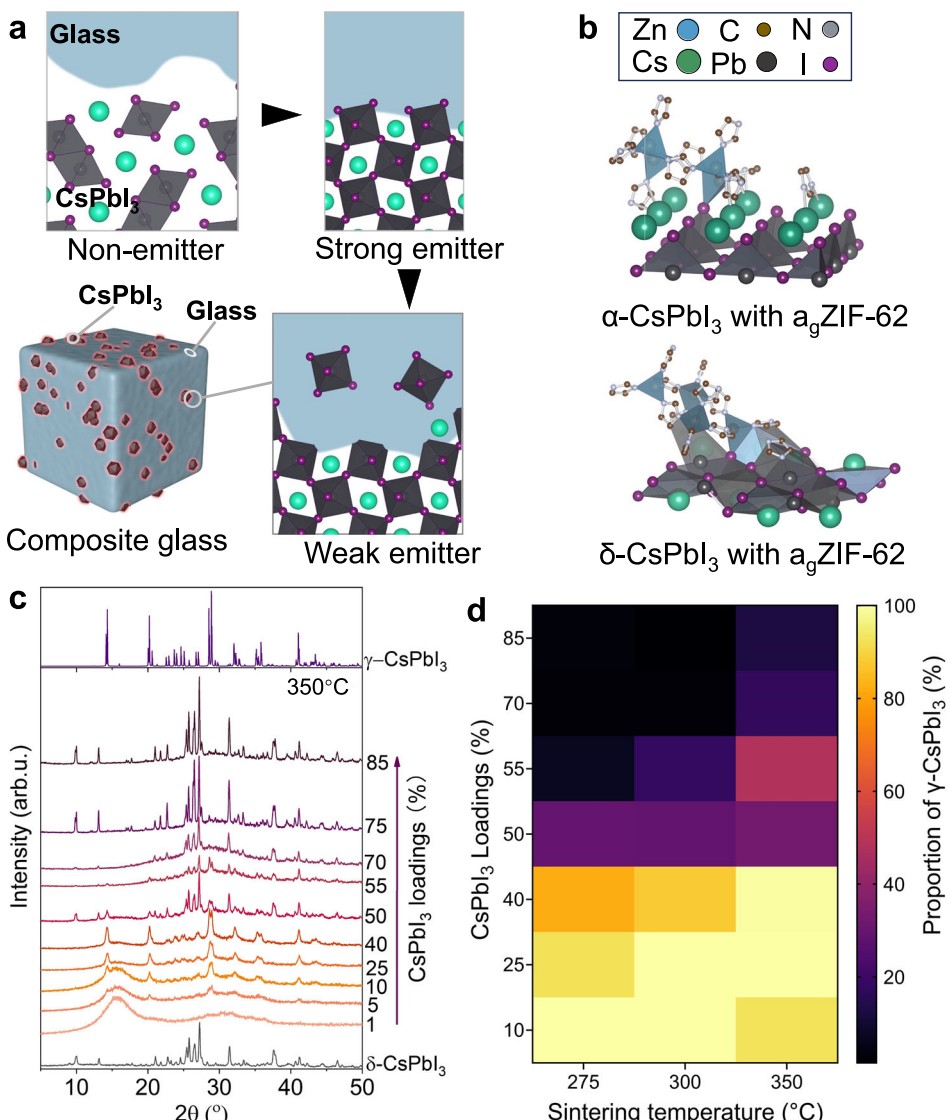

**Fig. 1 | Fabrication of (CsPbI₃)ₓ(a_gZIF-62)_Y composites. a** Schematic diagram of the CsPbI₃ phase transition and evolution of the interfacial atomic structures during sintering. Arrows indicate the progress of sintering. **b** Schematic diagram of the DFT calculation for composites with different CsPbI₃ crystal phases. **c** Ex situ XRD pattern of (CsPbI₃)ₓ(a_gZIF-62)_Y composites sintered at 350 °C. X-ray λ: 1.5406 Å. **d** γ-phase CsPbI₃ proportion within of different (CsPbI₃)ₓ(a_gZIF-62)_Y composites, as retrieved from Rietveld refinement of the XRD profiles.

maximum for sintering at 300 °C whereas (CsPbI₃)_{0.85}(a_gZIF-62)_{0.15} showed a PL maximum for sintering at 225 °C. Similar behaviour was also identified for the PL quantum yield (PLQY), with the highest PLQY of 81.3% for (CsPbI₃)_{0.05}(a_gZIF-62)_{0.95} at 275 °C sintering (Fig. 2a). This value is greater than the majority of previously reported CsPbI₃[29–33]. With increased CsPbI₃ loading, the PL peak red-shifted first and then blue-shifted (Supplementary Fig. 9), which was paralleled by similar changes in the γ-phase absorption edge shown in the UV–Vis absorption spectra (Supplementary Fig. 10). The composites demonstrated photostability under continuous excitation, showing negligible shifts in the peak position and variation in intensity during the test (Fig. 2b).

Scanning transmission electron microscopy (STEM) based cathodoluminescence (CL) further demonstrated that the CsPbI₃ nanometre-scale particles were indeed the source of the light emitted, and the luminescence properties of the composite exhibited different characteristics at the bulk nanocrystals and the interface regions (Fig. 2c–f). The emission spectra from these particles peaked at around 710 nm (Fig. 2e), and the inter-particle emission peak wavelength varied in the +/−5 nm range based on a Gaussian model fitting of the

dataset (Supplementary Figs. 11–12). In this field of view, selected to highlight a wide distribution of particles while still resolving individual grains, the variation of the emission intensity was not directly related to the size of the particles, indicating the quantum confinement was not the main contributor to the strong light emission. The width parameter σ of the Gaussian fitted to the emission spectra had a tendency for larger value at the edge of the particles (Fig. 2f). Histograms of the fitted coefficients for the data acquired across a relatively large (6.9 μm²) projected surface showed that the sample was quite homogeneous (Supplementary Fig. 12). In addition, correlations between emission peak centre wavelength with its width and intensity were observed (2D histograms in Supplementary Fig. 12). It is worth mentioning that to achieve single particle emission profile STEM lamellae samples were prepared by focused ion beam (FIB), which is known to be prone to damaging perovskites and to quenching CL emission. Some nanocrystals visible in the STEM image, particularly those at the surfaces of the lamella, are accordingly not expected to show strong emission. However, the FIB lamellae emitted light at the same wavelength and with the same emission spectral shape as those powder

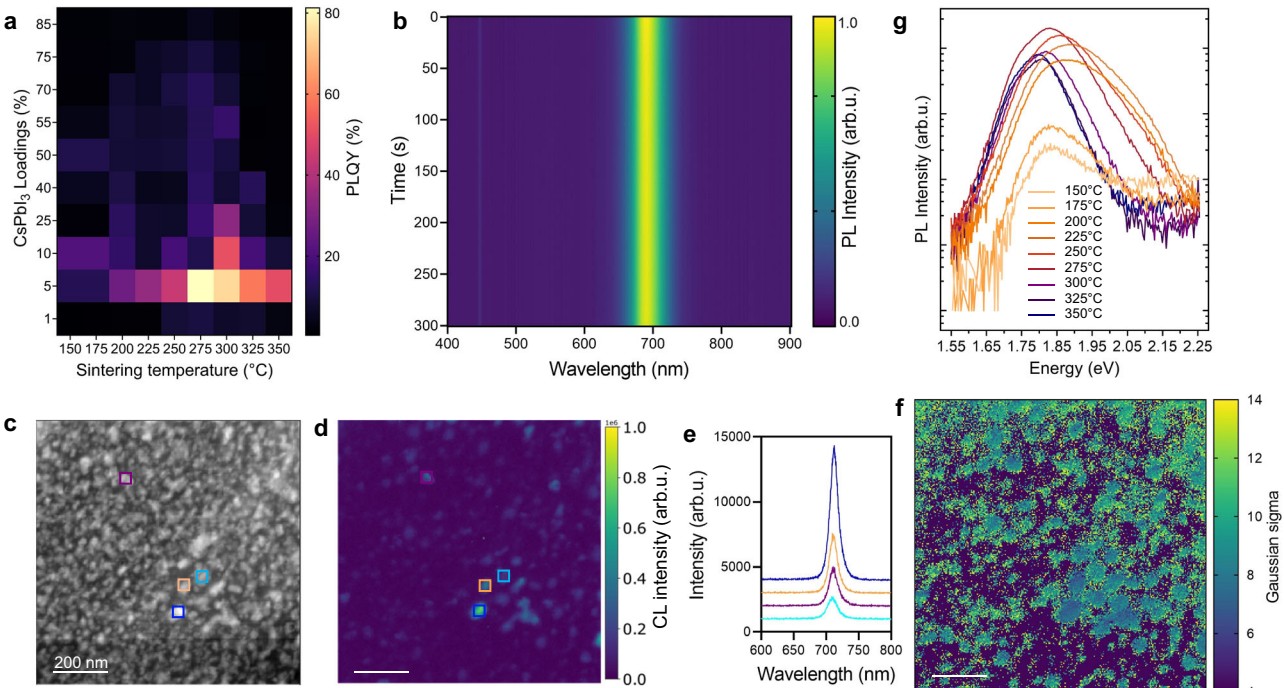

**Fig. 2 | Light emission profile of the composites. a** PLQY of different $(CsPbI_3)_X(a_gZIF\text{-}62)_Y$ composites. **b** Evolution of PL spectra under constant laser excitation (440 nm with 82 mW/cm²) for $(CsPbI_3)_{0.05}(a_gZIF\text{-}62)_{0.95}$. **c** High angle annular dark field (HAADF) image, **d** integrated CL intensity map around the main emission peak at 710 nm and **e** selected CL spectra in the regions of interest marked in (**c**) and (**d**) panel. **f** represents the Gaussian fitted σ (nm) values. Scale bar, 200 nm. **g** PL spectra for $(CsPbI_3)_{0.40}(a_gZIF\text{-}62)_{0.60}$ composite shown in log scale.

samples prepared by crushing the composite using a mortar and a pestle, confirming the strong stabilisation effects within the composites.

The PL emission properties from pure perovskites are closely related to their underlying crystal structures and trap states. Enhanced crystallinity should reduce bulk and surface defects, suppressing non-radiative exciton recombination and enhancing PL emission[34,35]. The data presented here, on the other hand, clearly illustrates the presence of additional prominent effects within the composites, such as the fact that a higher sintering temperature (e.g., >300 °C) enhances crystallinity of the active $CsPbI_3$ but dramatically reduces the PL for all composites. The PL spectra of $(CsPbI_3)_{0.40}(a_gZIF\text{-}62)_{0.60}$ is plotted on a log scale against photon energy to elucidate the nature of the trap states for the composites (Fig. 2g and Supplementary Fig. 13). PL emissions consist of a primary Gaussian-type peak and can contain tails on the low-energy shoulders usually being associated with the formation of near-edge traps[36]. For the case of $CsPbI_3$ perovskite, the lower-energy Urbach tail is typically associated with disorder at the crystal surface[37,38].The Urbach tail slopes for composites remained generally unchanged as sintering temperature increased, demonstrating that the defects at the crystal surface can be well passivated even at the early stages of sintering. This is consistent with the fitting of PL spectra with a two-Gaussian function (Supplementary Figs. 14, 15)[39], in which the low energy peaks contributed only a minor to modest part of the overall intensity.

## Composite property evolution during sintering

Differential scanning calorimetry (DSC) can reveal overall information on the thermodynamic and kinetic phase transitions, interfacial interactions and particle aggregations during sintering. DSC curves of pure $CsPbI_3$ and $a_gZIF\text{-}62$ were relatively featureless except for their phase transition (δ- to α-$CsPbI_3$, ca. 320 °C) and $T_g$ ($a_gZIF\text{-}62$, *ca.* 290 °C) (Supplementary Fig. 1 and Supplementary Fig. 16a). In contrast, a broad endothermal peak for the composites began to develop

at temperatures as low as 110 °C (Fig. 3a) during the first heating ramp (with negligible weight loss, Supplementary Fig. 16b), and the cooling and 2nd heating ramp were featureless, except for the $T_g$. The endothermal enthalpy of the composite (ca. from 110 to 190 °C) was much greater and wider than the phase transition enthalpy of the corresponding $CsPbI_3$. Furthermore, the $T_g$ of the composite decreased with higher $CsPbI_3$ loading, implying that the compositing increased entropy and changed the chemical composition of the $a_gZIF\text{-}62$ phase (Fig. 3b and Supplementary Fig. 17).

Particle coarsening and the evolution of a diffusion layer at the interface were identified via the synchrotron X-ray small angle scattering (SAXS) on $(CsPbI_3)_{0.1}(a_gZIF\text{-}62)_{0.9}$ composites (Fig. 3c and Supplementary Fig. 18). Within the measured scattering vector ($q$) range, higher temperature sintering coarsened small $CsPbI_3$ particles particularly when temperature approached the $T_g$ (275 °C), where the viscous flow rates of MOF liquid facilitated the diffusion of the perovskite particles. This also suggested the DSC endothermic feature at ca. 110–190 °C was predominantly attributed to phase transition and interfacial interaction, instead of particle coarsening. Further analysis of the SAXS with the Porod's plot (Fig. 3d) showed higher sintering temperature led to more significant positive deviation from the Porod's law[40], i.e., $\lim q^4 I(q) = K$ when $q \to \infty$, where $q$ is the scattering vector, $I(q)$ is the scattering intensity and $K$ is the Porod constant. The Porod's law describes a system with sharp phase boundary with clear contrast of electron density. The positive deviation may originate in the micro-fluctuations of electron density within the sample, indicating the evolution of a diffusion layer along with sintering.

The development of interfacial bonding within the composites was studied with in situ temperature-resolved synchrotron terahertz (THz) FarIR vibrational spectroscopy (Supplementary Fig. 19). The primary characteristics of $(CsPbI_3)(a_gZIF\text{-}62)(10/90)$ can be attributed to Zn-N stretching (ca. 300 cm⁻¹) and the aromatic ring deformation (ca. 670 cm⁻¹) (Supplementary Fig. 20)[41]. Additional features (*ca.* 135 and 275 cm⁻¹) associated with Zn-I formation emerged from ca. 110 °C

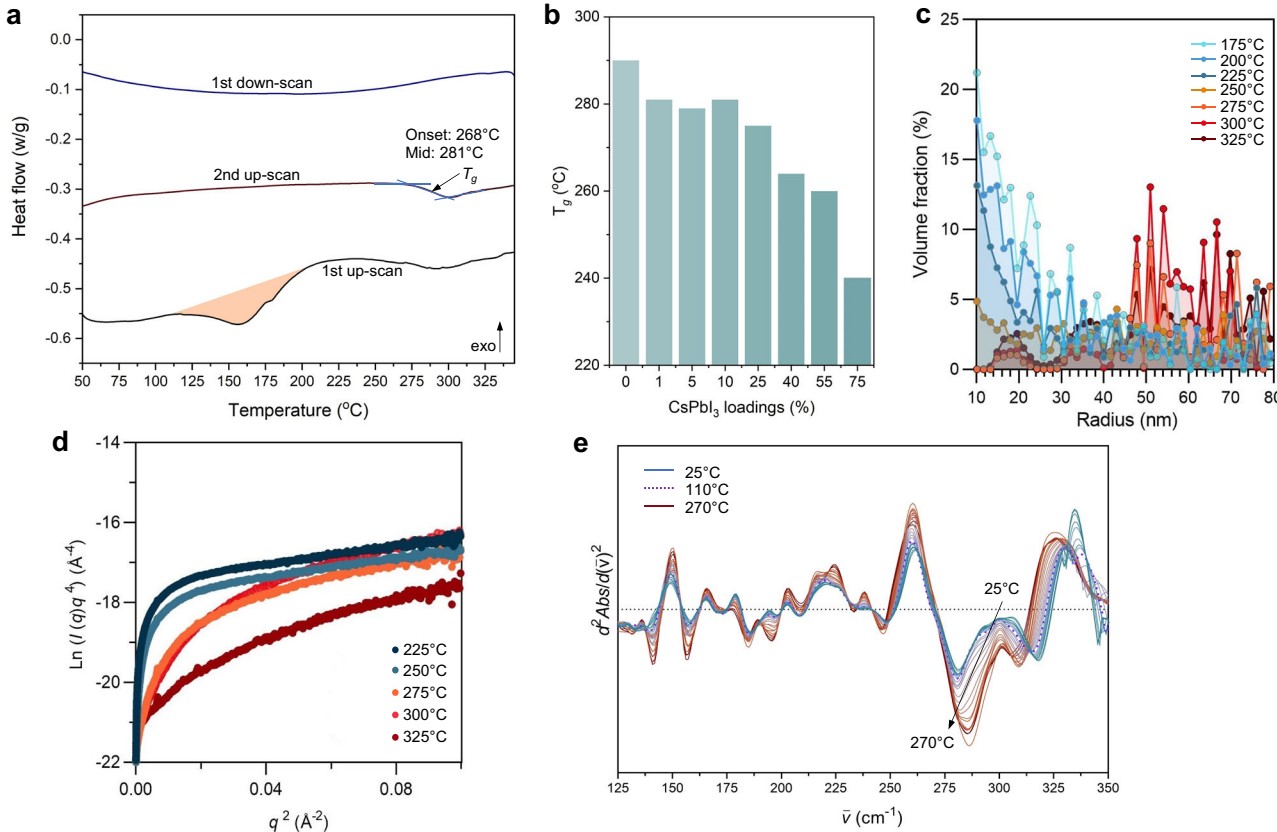

**Fig. 3 | Evolution of interfacial interaction during the sintering. a** DSC results for $(CsPbI_3)(a_gZIF-62)(10/90)$. Data was collected under constant flowing nitrogen protection (20 mL/min). The temperature ramping rate for the first up-scan was 20 °C /min, and the ramping rate was 10 °C /min during the DSC cooling and second up-scan. **b** Glass transition temperature, $T_g$, of the composites as a function of

CsPbI₃ loading. **c** Particle size evolution for $(CsPbI_3)_{0.1}(a_gZIF-62)_{0.9}$ composites sintered at different temperature. **d** Change of the Porod's plot of the corresponding SAXS patterns. **e** Second derivative of temperature-resolved in situ THz FarIR spectra for $(CsPbI_3)(a_gZIF-62)(10/90)$ composites.

(concurrent with the endothermal feature in DSC), visible in the 2nd derivative of the in situ spectra (Fig. 3e). With further increase of the sintering temperature, the Zn-I features gradually intensified and remained substantially invariant after cooling. Similar behaviour was found in other composite samples (Supplementary Figs. 21–24), and the variation of bonding environment was confirmed by X-ray photoelectron spectroscopy (XPS) (Supplementary Figs. 25–26).

### Element specific atomic-scale structures at the interface

Solid state nuclear magnetic resonance (SS-NMR) spectroscopy can shed a clear picture on the Cs chemical environment evolution. It identified a nice interplay between different perovskite phases and the greatest proportional contribution of interfacial Cs occurred at 250 °C sintering, correlating with the fluctuation in PL intensity against different sintering temperature. The disordered nature of the glassy matrix was confirmed by the broad [1]H and [13]C NMR spectra (Supplementary Fig. 27)[42]. In the [133]Cs NMR spectra (Fig. 4a), for the sample without sintering, the narrow peaks at ca. 260 ppm and 280 ppm were attributed to δ-CsPbI₃ and CsI, respectively. The broad, low-amplitude signals extending between 0 and ca. 350 ppm were attributed to poorly crystalline, highly defective CsPbI₃, as expected for mechanochemically synthesised perovskites[26]. After sintering, the broad contributions and CsI peaks diminished, and the major signals stemmed from γ-CsPbI₃. Notably, the asymmetric 'shoulder' of the γ-CsPbI₃ peaks (below ca. 160 ppm, Fig. 4a) was related to the γ-CsPbI₃ impacted by the chemical environment at the interface, directly in contact with glassy matrix, forming gradient interdiffusion and modified bonding at the interface. In addition, [1]H-[133]Cs cross-polarization magic angle

spinning (CPMAS) NMR showed a non-negligible cross-polarization transfer between the protons of $a_gZIF-62$ and the "shoulder-peak" of γ-CsPbI₃ at the interface (Fig. 4b), pointing to close proximity between the imidazole ligands and Cs atoms. This interfacial Cs contribution was the most substantial for the composite sintered at 250 °C, and gradually decreased along with higher sintering temperature and particle coarsening, showing similar tendency to the variation of PL intensity against different sintering temperatures.

Extended X-ray absorption fine structure (EXAFS) and wavelet transforms (WTs) were performed on to explore the change of Zn and Pb coordination environments through sintering (Supplementary Figs. 28–29), and to identify the change of the first layer structure for these atoms. Wavelet transformation (WT) provides complementary information to FT (Fourier transform)-EXAFS, which can help differentiate different coordinating pairs at similar atomic distances. EXAFS spectra and wavelet transform contours of the Zn K edge (Fig. 4c–e and Supplementary Fig. 29) revealed both samples exhibited main peaks at the R space distance of ca. 1.5 Å, corresponding to the distance of Zn-N pairs. The local environment of the Zn prior to sintering was identical in the crystalline ZIF-62[43], indicating Zn atoms maintains their tetrahedral configuration within the composites. After sintering, the emerging peak at ca. 2.7 Å can be assigned to Zn-I, in good agreement with the THz results, which are assigned to Zn ions taking the place of $V_{Cs}$ (Cs vacancy), as it has the lowest transition level among all kind of defects[44]. Change to the Pb local environment were more pronounced, particularly considering the evolution of disparate crystal structures (Fig. 4f–h and Supplementary Fig. 30). The peaks prior to sintering located at *ca.* 1.7 Å and 2.8 Å can be assigned to Pb-O and Pb-I,

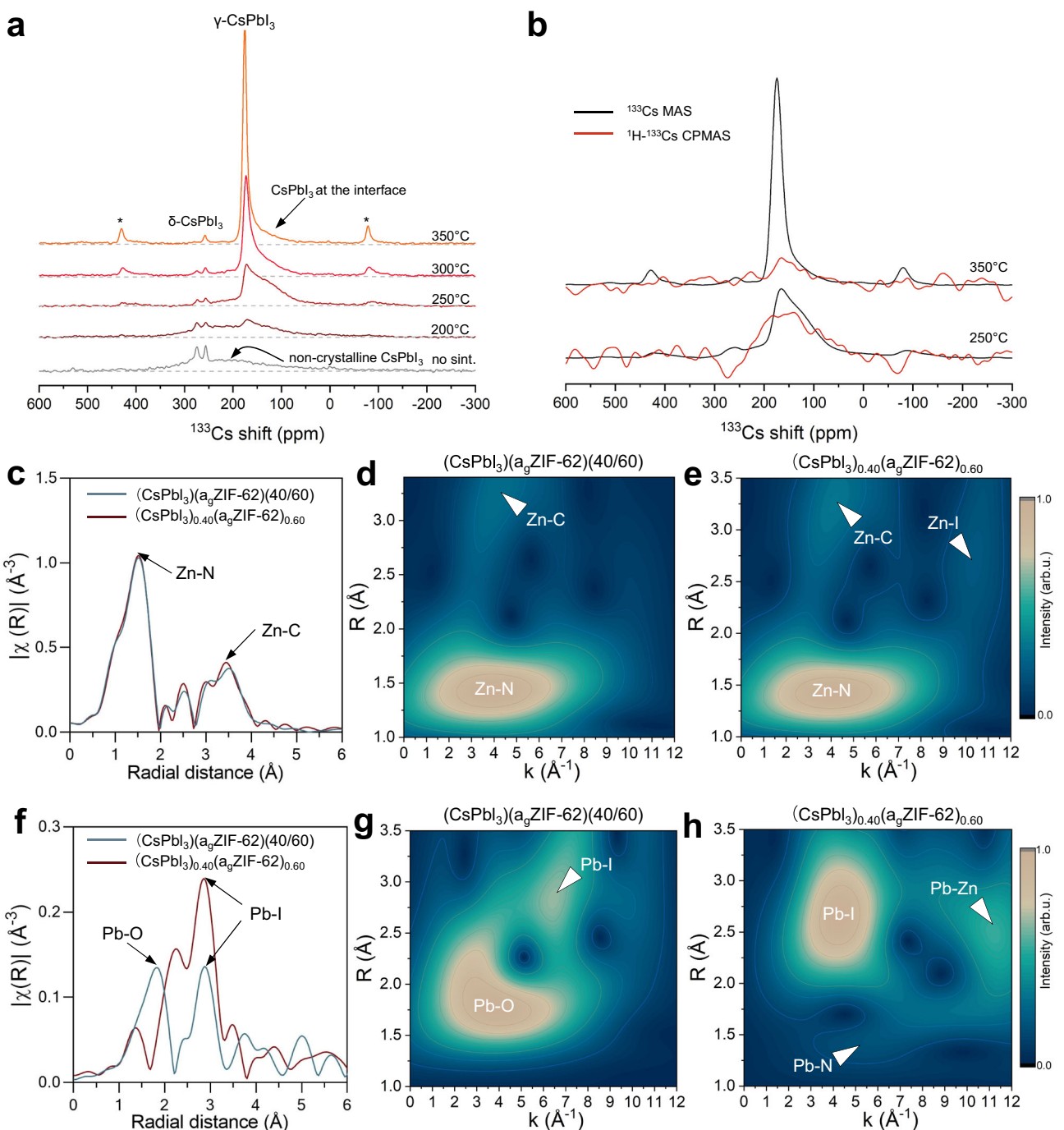

**Fig. 4 | Structure of the interface. a** $^{133}$Cs MAS NMR spectra of $(CsPbI_3)_{0.25}(a_gZIF-62)_{0.75}$ composites sintered at 200, 250, 300, and 350 °C. **b** $^1H-^{133}Cs$ CPMAS NMR spectra of $(CsPbI_3)_{0.25}(a_gZIF-62)_{0.75}$ sintered at 250 and 350 °C. **c–e** Extended X-ray absorption fine structure (EXAFS) signal and the full-range wavelet transform (WT) representation for the Zn K edge of $(CsPbI_3)_{0.40}(a_gZIF-62)_{0.60}$ composites. **f–h** Extended X-ray absorption fine structure (EXAFS) signal and the full-range wavelet transform (WT) representation for the Pb $L_3$ edge of $(CsPbI_3)_{0.40}(a_gZIF-62)_{0.60}$ composites. Composites were sintered at 350 °C.

respectively[45]. The presence of Pb-O is attributed to the surface oxidation through high energy ball milling synthesis. Crystallisation of CsPbI$_3$ during sintering eliminate the Pb-O bonding and, therefore, Pb-I became more dominating after sintering (Fitting parameters shown in Supplementary Table 1). In addition, the emerging peak at *ca.* 1.4 Å and 2.5 Å can be attributed to Pb-N and Pb-Zn pairs formed at the interfaces. The formation of new atom pairs was supported by the X-ray adsorption near-edge structure (XANES) of the Pb L$_3$-edge spectra (Supplementary Figs. 28–29), which exhibit a shift toward lower energy after sintering, due to the higher electronegativity of N and Pb

over Zn. The microscopic mechanism of ZIF melting involves the breaking and reformation of Zn-N bonds, which permits Pb ions to form coordinate with imidazolate-termination sites, resulting in new Pb-N bonds. Meanwhile, while a minor portion of Pb ions interacted with Zn-termination or Zn ion replaces V$_I$ (I vacancy) from the diffused alloying layer forming Pb-Zn[46]. Furthermore, all of the peak intensities in the $(CsPbI_3)_{0.25}(a_gZIF-62)_{0.75}$ EXAFS plot are higher than those of $(CsPbI_3)_{0.40}(a_gZIF-62)_{0.60}$, indicating that the $(CsPbI_3)_{0.40}(a_gZIF-62)_{0.60}$ contains more defective sites, which is consistent with corresponding their PLQY results (2.17% vs 1.23%).

Annular dark-field scanning transmission electron microscopy (ADF-STEM) identified the emergence of interdiffusion alloying layers after high temperature sintering. FIB milling was first used to extract a *ca.* 10 nm thick lamella from a piece of $(CsPbI_3)_{0.1}(a_gZIF-62)_{0.9}$ sintered at 350 °C to expose individual LHP grains (Supplementary Fig. 31). The sample showed pronounced atomic number contrast between two phases. In STEM, electron energy loss spectroscopy (EELS) provides a sensitive method for analysing the chemical environment using transmitted sub-nanometre electron probes[47]. Although we cannot rule out entirely contributions from FIB milling-induced modifications to the sample, after 350 °C sintering, an EELS line scan across the interface quantifies an excess quantity of I outside the $CsPbI_3$ (Fig. 5a, b and Supplementary Fig. 32), while such a behaviour was not observed at 275 °C sintering (Supplementary Fig. 33). The atomic structure and spatial variation in elemental composition was further corroborated by ADF-STEM imaging and STEM-based energy dispersive spectroscopy (STEM-EDS) (Supplementary Figs. 34–35), showing an abrupt termination of the crystals surrounded by a non-crystalline boundary consistent with prior diffraction experiments[26] and a similar extent of I outside of the $CsPbI_3$ crystals. The gradual diffusion of heavier I atoms towards the glassy phase at high temperature aligns with the micro-fluctuation in electron density as demonstrated in SAXS, and the gradual intensified Zn-I THz features up on sintering. The diffusion of Cs, in comparison, has not reached beyond nanometre range from the perovskite grain.

Atomic pair distribution functions (PDFs), a technique which unavoidably measures the average structure of the whole system. In this work, two physical features aid us to differ the interface from the bulk; (i) a relatively high surface to volume ratio in the target crystal beneficially weights interface-related signals, and (ii) there exists clear contrast between the short-range fine structure existing at the alloying layer compared to the long-range ordering in the bulk, with the alloying layer being heavily disordered, deviating from the prominent periodic fine structure found in the bulk. This analysis identified the interfacial diffusion layers mainly contained Pb-I, Cs-I and Pb-N atomic pairs with only short-range orders. Ex-situ synchrotron X-ray total scattering measurements were performed on a series of composites sintered at 350 °C. The PDF, obtained by Fourier transform, showed the short-to-long range fine structure of the composites (Fig. 5c). Rietveld analysis of the composite $(CsPbI_3)_{0.50}(a_gZIF-62)_{0.50}$ was performed with desolvated ZIF-62, γ- and δ-$CsPbI_3$ reference structures. We found that ZIF structure was only maintained in the short-range order. Good fitting was achieved in the medium and long-range ($R_{wp}$ of 11 % in the range of 15–50 Å), with 13.7 wt% of γ-$CsPbI_3$ and 30.1 wt% of δ-$CsPbI_3$ being identified. The difference pattern in the short-range suggested the structural distortion originated from the surface of the crystalline phases, with the main feature being assigned to Pb-I (ca. 3.2 Å). This is an indication of presence of discrete, rigid and slightly distorted Pb-I octahedra within the diffusion alloying layer. These

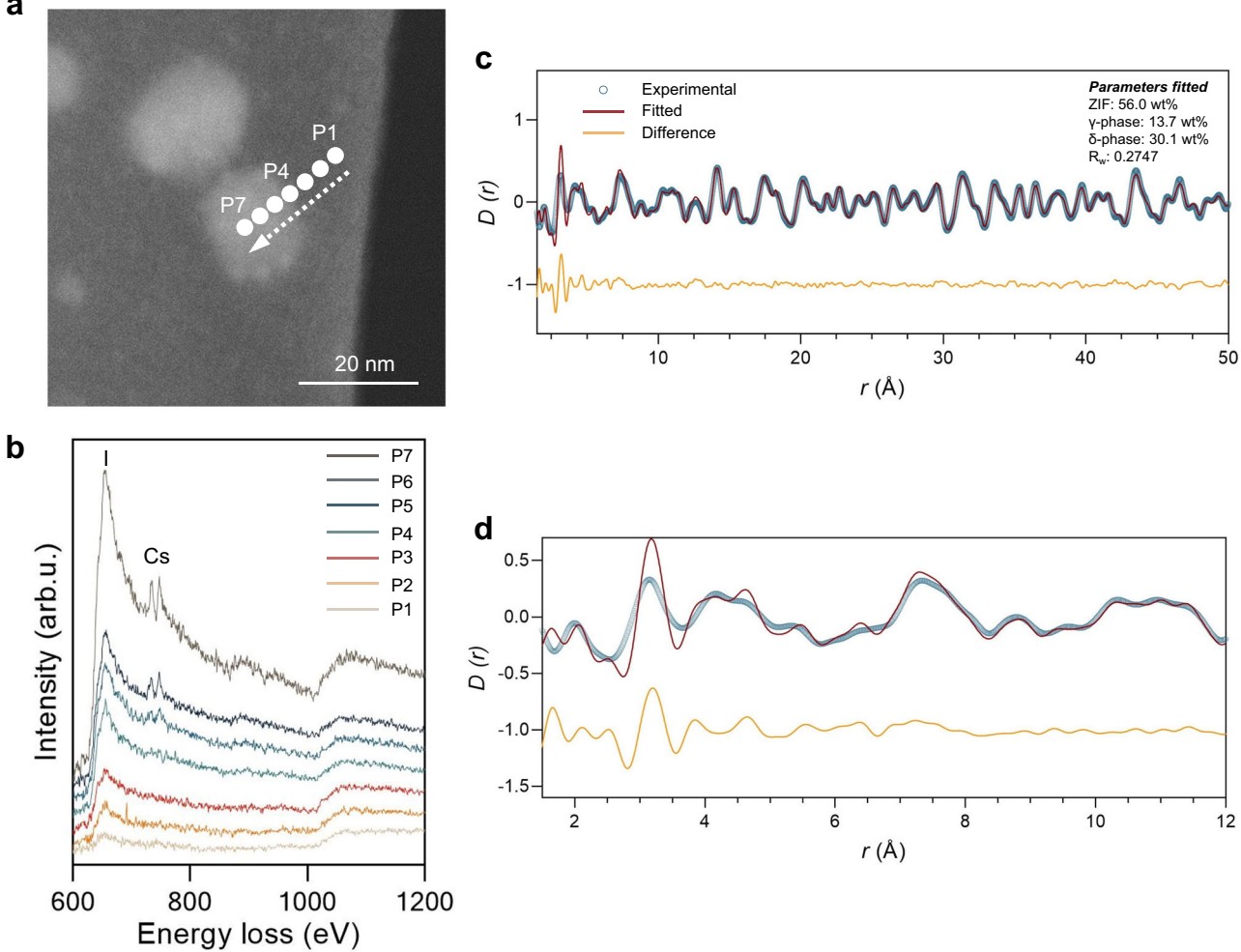

**Fig. 5 | Chemistry and structure of the alloying layer at the interface. a** ADF-STEM image of $(CsPbI_3)_{0.1}(a_gZIF-62)_{0.9}$ composites sintered at 350 °C. **b** EELS core loss spectra acquired from the marked points shown in TEM image with background subtracted. **c, d** Atomic pair distribution functions for $(CsPbI_3)_{0.50}(a_gZIF-62)_{0.50}$ composites sintered at 350 °C. Pair distribution function *D(r)* calculated via Fourier transform of the X-ray total scattering function. Panel (**d**) is the enlarged fitted data in the r range of 1.5–12 Å.

difference features are highly composition- and sintering temperature dependent, showing more contribution from these amorphous perovskite components at higher sintering temperatures, or from composites with larger expected interfaces (Supplementary Figs. 36–37). These observations further confirm the amorphous structures are originated from the interface, instead of the amorphous regions of the bulk perovskites.

## Discussion

This study identified the presence of interfacial alloying layers between perovskite and hybrid glass phases, which were mainly consisted of amorphous Pb-I structures. Different behaviours were observed for different atoms during the alloying: high temperature sintering promoted the crystallisation for Cs to form bulk perovskites, possible through a fluxing mechanism. In comparison, the diffusion of the I into the glassy phases was significant (can reach up to ca. 10 nm away from the $CsPbI_3$ grain), together with the preservation of Pb-I octahedron structure within the diffused alloying layer.

This work also unveiled the complex performance–property relationship within the $CsPbI_3$ glass composites. We identified that the optoelectronic performance of $CsPbI_3$ composites was predominately determined by the interface, instead of the bulk particle crystallinity. The interfacial alloying favoured the formation of active $CsPbI_3$ phases, stabilised the optoelectronic performance and, more importantly, effectively passivated the trap states. Nevertheless, it can also be detrimental to light emission: for all composites with varying $CsPbI_3$ loadings, the maximum PL intensity was obtained by sintering at a temperature at the corresponding $T_g$. Above the $T_g$, glass enters a viscous flowing state, leading to faster diffusion and generate non-stoichiometric perovskite near the interface. Though the coarsening can promote the overall crystallinity, the diffusion of Pb-I away from $CsPbI_3$ can lead to non-radiative recombination and quenching of PL. The principle can be expanded to other ABX perovskite systems because they share a common optoelectronic backbone of metal-halide orbitals.

This study has notable implications for the fundamental understanding of defect formation and control in perovskite materials and the design of halide perovskite devices through interfacial engineering. It exemplified that how the interface can passivate and then generate defects. The investigation of this multimodal system has provided significant progress in comprehending the fundamental processes and interfacial atomic level structures, shedding light onto governing the performance of perovskite materials in practical applications such as solar panels, LEDs and sensing devices. In particular, the notable stability under UV irradiation and relatively high temperature allows these composites to be further processed by an ultrafast laser. The composites generated a localised liquid phase as a result of the intense thermal accumulation, which can be used to fabricate optical products by three-dimensional printing. The whole process can be adjusted by optimising the pulse duration, repetition rate, and pulse energy as well as ultrafast-laser irradiation time. Furthermore, some advanced fabrication methods like chemical vapour deposition and atomic layer deposition, are promising to reduce the thickness of MOF glass to nano-metre size, making these composites more compatible with planar in device components. It is important to keep in mind that when using these composites in real-world applications, the material's limitations should never be disregarded. For instance, the ZIF glass is prone to chemical attack by acid and strong chelating agents' chemical attacks and can begin to decompose at a temperature of 600 °C. This indicates that when facing the mentioned problem in the design of practical devices, composites may require further encapsulation. Our findings indicate that prioritising the attainment of bulk crystal perfection may not necessarily lead to the most advantageous outcomes for improving the performance of this particular class of semiconductors. Rather, further research efforts should be focused on interfacial engineering, with consideration of different diffusion behaviour for different perovskite components.

## Methods

### Preparation of crystal ZIF-62 and $a_g$ZIF-62

Crystalline ZIF-62 was synthesised by hydrothermal methods. For ZIF-62-bim$_{0.05}$ solvothermal reactions, 1.2 g $Zn(NO_3)_2$, 891 mg imidazole and 12 mg benzimidazole were dissolved in 90 mL N,N-dimethylformamide (DMF), the whole solution was transferred in a 120 °C oven for 5 days. Off-stoichiometry ZIF-62 was used in this work to lower $T_g$. After cooling to room temperature, the solution was centrifugated to get crystal solid. To remove the unreacted precursors, a three-time washing-redispersion cycle in DMF was applied. After that, the product was dried in a 150 °C oven overnight[48].

ZIF-62 glass ($a_g$ZIF-62) was obtained by heating crystal ZIF-62 in a tube furnace with a ramping rate of 20 °C/min. Once the temperature reached 450 °C, the whole tube was immediately removed from the furnace and quenched in liquid nitrogen. The whole process was under argon protection until the sample cooled down to room temperature.

### Fabrication of $CsPbI_3$ perovskite

Inorganic lead halide perovskite ($CsPbI_3$) was prepared by a solvent-free mechanochemical process. A total amount of 400 mg stochiometric mixture, which was composed of CsI (144.17 mg) and $PbI_2$ (255.83 mg), as well as zirconium oxide balls (16 g) were added to a ball milling jar, then the ball milling was performed with a planetary ball milling machine at 800 RPM for 1 h. After that, the solid product was recovered, and then stored under ambient conditions prior to further testing.

### Preparation of $(CsPbI_3)_X(a_g$ZIF-62$)_Y$

Powder $(CsPbI_3)_X(a_g$ZIF-62$)_Y$ samples were prepared by mixing a total 200 mg of $CsPbI_3$ and agZIF-62 via ball milling (16 g balls) at 800RPM for 1 h, here $X$ and $Y$ refer to mass ratios. For example, 20 mg $CsPbI_3$ perovskite mixed with 180 mg $a_g$ZIF-62, the resultant mixtures were referred to as $(CsPbI_3)_{0.1}(a_g$ZIF-62$)_{0.9}$, where 0.1 and 0.9 are the mass percentage of each component. For all samples in this search, $X$ values changed from 0.01 to 0.85. The fabricated powder mixture samples were transferred to a tube furnace for thermal treatment and quenching process, which was similar to the ZIF glass preparation process. The thermal treatment temperature ranged from 150 to 350 °C.

### Scanning electron microscopy (SEM)

The surface morphologies and atomic density information near the surface were investigated with a high-resolution scanning electron microscope, JEOL JSM-7100F, under both secondary electron and backscattering mode (15.0 kV acceleration voltage). All samples were dried under an 80 °C vacuum oven for eight hours, followed by a surface platinum coating. All samples were applied brief plasma surface cleaning before SEM analysis.

### Focused ion beam scanning electron microscopy (FIB-SEM)

An FEI SCIOS focused ion beam scanning electron microscope (FIB/SEM) dual beam system was employed to generate TEM lamella. The area of interest on the sample surface was coated with a 1 μm thick layer of platinum to shield it from ion bombardment damage. Subsequently, applying a Ga ion beam current ranging from 3 to 30 nA at an acceleration voltage of 30 kV, a thin slice of the material was removed perpendicularly to the sample surface by milling two trenches on both sides. After being detached from the sample matrix, the thin segment was bonded on a TEM half-grid after being mounted to the EazyLift Micromanipulator. Following post-thinning with a significantly reduced ion beam current (from 0.3 nA down to 30 pA), low voltage polishing at grazing incidence was used to complete the cleaning process.

## Powder X-ray diffraction (XRD) analysis

Room temperature powder XRD analysis was collected with a Bruker D8 Advance MKII diffractometer using Cu Kα radiation with divergent (Bragg-Brentano) geometries. The 2θ range was 5 to 50°, with a step size of 0.02° and a step rate of 10 s. The temperature resolved in situ powder XRD was performed by Rigaku Miniflex 600 Benchtop X-Ray Diffractometer equipped with Anton Paar XRD900 furnace with nitrogen protection. The ramping rate was 20°C/min, with the 2θ range of 5–50°, a step size of 0.05° and a step rate of 10 s. The measurement was performed under constant flow of nitrogen.

## Synchrotron powder XRD analysis

In situ synchrotron x-ray powder diffraction data was collected at PETRA III beamline P21.1 at the Deutsches Elektronen-Synchrotron (DESY), Hamburg, Germany. For high-resolution XRD, a defocused x-ray beam (101 keV, $\lambda = 0.10173$ Å) was applied, with detector distance of 1950 mm. During the measurement, a steady flow of nitrogen was introduced into a 1.3 mm quartz capillary containing the sample. The sample was heated with Linkam TS1500 at a ramping rate of 20 °C/min to 350 °C and cooled back to room temperature after an isotherm stage for -100 s at the highest temperature. To keep the phase fractions constant, the diffraction intensities were normalised to the integrated signal at high angles.

## Thermogravimetric and calorimetric analysis

Thermogravimetric analysis (TGA) was conducted using a Mettler Toledo. The sample was placed in an alumina crucible and heated at a ramping rate of 20 °C/min under flowing (20 mL/min) Ar environment.

Mettler Toledo differential scanning calorimetry (DSC) 1 STARe system was used to determine the calorimetric features. For each measurement, $ca.$ 10 mg sample was loaded into an aluminium crucible and heated under a flowing Ar (20 mL/min) environment. To determine the melting ($T_m$) and thermal decomposition ($T_d$) and crystal phase transaction temperatures, samples were heated at a ramping rate of 20 °C/min to the targeted temperature. To determine the glass transition temperature ($T_g$), samples were heated above their melting temperature at 20 °C/min, and then cooled back to 40 °C at 10 °C/min, and then ramped up at a rate of 10 °C/min in the second upscan.

## UV−Vis absorption

To determine the bandgap ($E_g$) changes of crystal ZIF-62, $a_g$ZIF-62, powder mixture and ($a_g$ZIF/perovskites) composite pellets, UV−Vis absorption spectroscopy was carried out using a Jasco V-650 UV−Vis spectrophotometer across a wavelength range of 400 nm to 800 nm. The bandgap was determined according to the equation of $E_g = 1240/\lambda_g$, where $\lambda_g$ refers to the adsorption wavelength of different samples.

## Photoluminescence measurements (PL)

Photoluminescence (PL) and time-resolved fluorescence emission spectra were collected at room temperature on an Edinburgh Instrument FLSP-900 fluorescence spectrophotometer. The steady PL spectra were measured with a monochromatized Xe lamp light source, while the time-resolved PL (TRPL) decays were measured with a 375 nm pulsed diode laser excitation source. Photoluminescence quantum yield (PLQY) of the samples were measured using FS5 fluorescence spectrometer (Edinburgh Instruments). Absolute PLQY measurements were performed in a pre-calibrated integrating sphere. The samples were excited at 440 nm. The photostability was measured with constant laser excitation. The excitation spot was near-perfect circular (beam passed through the circular iris) with diameter -0.8 mm. The excitation wavelength was 440 nm.

## Synchrotron Tera-Hz Far-Infrared (THz/Far-IR) absorption spectroscopy

THz/Far-IR absorption spectra were acquired using a Bruker IFS 125/HR Fourier Transform (FT) spectrometer at the Australian Synchrotron's THz/Far-IR beamline. To increase the signal-to-noise ratio, the bolometer was kept in cryogenic mode with liquid helium, and a 6 m thick Multilayer Mylar beamsplitter was utilised.

Attenuated total reflection (ATR) was used for the measurement. Pressure was used to hold samples in place on the surface of the diamond crystal window. The ATR heating stage was used to capture temperature resolved in situ spectra, and the sample was held under flowing Ar (about 20 mL/min). For data processing, the Extended ATR correction algorithms in the OPUS 8.0 programme were used, along with the NumPy module v1.15 and Python v3.5, for spectral data correction and peak fitting[49].

## X-ray absorption spectroscopy and pair distribution function

XAS measurements at the Zn K-edge and Pb L-edge were performed at the 10-ID-B at APS beamline of Argonne National Laboratory. Data was collected from 100 eV below the Zn K-edge to ~600 eV above the edge, and 100 eV below the Pb L-edge to ~800 eV above the edge. Samples were loaded into 0.0395″ inner diameter thin-walled Kapton X-ray capillaries and examined in fluorescence mode. Data processing and subsequent modelling was performed using the Demeter XAS software package. Zn K-edge and Pb L-edge EXAFS fitting was performed by background normalization, k2 weighting, and Fourier transform. The Zn-N and Zn-C shells from the reported ZIF-62 structure were employed for the fitting of Zn K-edge. And the Pb-O, Pb-I, Pb-Zn shells were obtained from the DFT modelling of $Pb_3O_4$, $CsPbI_3$, and $ZnPb_3$[50–52].

The distance and peak broadening were calibrated using $CeO_2$ NIST SRM 674b. Data were radially integrated in pyFAI without absorption correction, while PDFs were calculated with GSAS-II, and fit using PDFgui. PDFs were fit from 1.5 to 50 Å using previously determined single crystal structures for the ZIF, γ-, and δ-phase $CsPbI_3$. The refined lattice parameters of the $CsPbI_3$ were $a = 8.832$, $b = 12.485$, $c = 8.651$ and 10.457, $b = 4.795$, $c = 17.773$ for the γ and δ phases respectively.

For comparison, higher $Q_{max}$ (25 Å$^{-1}$) data was obtained from beamline 11-ID-B at the Advanced Photon Source at 58.6 keV for sample containing 40 wt% $CsPbI_3$. While larger $Q_{max}$ at shorter detector distances theoretically enhances the resolution of the PDF, in this case the sharp reflections of the crystalline $CsPbI_3$ phases become undersampled, leading to comparable PDF information. The beam sensitivity of the ZIF complicates the acquisition of high-resolution analyser crystal datasets.

## Transmission electron microscopy (TEM)

Electron energy loss spectroscopy (EELS) was acquired using a Hitachi HF5000 scanning transmission electron microscope (STEM) at the University of Queensland equipped with a Quantum ER Gatan Imaging Filter (GIF) and operated at 200 kV. EELS data were acquired using a dispersion of 0.5 eV/channel in dual EELS mode to obtain zero loss peak and core loss edge information in tandem for each probe position. Collection and convergence semi-angles were recorded as 29 mrad and 32 mrad, respectively.

EELS data were processed using HyperSpy[53], an open-source Python package for electron microscopy. First, the zero loss peak centring at 0 eV was checked. Background subtraction was carried out by power law fitting between 580 and 620 eV. Following background removal, spectra were cropped from 580 eV, the lowest energy used in the background fitting window before the I $M_{45}$ edge with a first onset expected at 619 eV energy loss[54]. Spectra without background subtraction were normalised by dividing by each spectrum by the total integrated core loss intensity (to remove effects from differences in total inelastic scattering).

## Cathodoluminescence (CL) in a STEM

Scanning transmission electron microscopy (STEM) imaging and CL spectroscopy were performed on a modified Nion Hermes200 operated with 100 keV kinetic energy electrons. With this microscope, a subnanometer electron beam can be generated for high spatial resolution imaging and spectroscopy[55]. High angle annular dark field (HAADF) images have an intensity which is proportional to the projected atomic number along the electron beam trajectory. Therefore, the heavier $CsPbI_3$ particles appear brighter in the lighter MOF. CL was performed with a Mönch system from Attolight fitted with a diffraction grating 150 grooves per millimetre (0.34 nm dispersion on the spectrometer used), which gives a wavelength resolution of about 1 nm at 600 nm[56]. The optical spectrometer was calibrated using an Ar-Hg lamp. The sample was kept at around 150 K using a liquid nitrogen cooled sample holder (HennyZ).

Additional ADF-STEM and STEM-based X-ray energy dispersive spectroscopy (STEM-EDS) were acquired using an FEI Titan[3] Themis (Thermo Fisher) equipped with a high-brightness 'X-FEG' electron source and a four-quadrant (0.7 sr) Super-X EDS detector (Bruker) and operated at 300 kV.

## X-ray photoelectron spectroscopy (XPS)

XPS data was acquired at room temperature on a Kratos AXIS Supra Plus XPS system which uses a dual monochromated Al Kα/Ag Lα X-ray source. At pass energies of 160 and 20 eV, the survey and high-resolution spectra of vacuum-dried samples were recorded, respectively. During sample analysis, the chamber's basal pressure was between $10^{-9}$ and $10^{-8}$ Torr. Duplicate scans were processed, and a single area on each sample was analysed. The software CasaXPS was employed to execute peak fitting on the high-resolution data. Charge correction was performed on all high-resolution scans to the C-C peak at 284.6 eV.

## Synchrotron X-ray small angle-angle scattering (SAXS)

Ex situ X-ray scattering data were gathered at the SAXS/WAXS beamline of the Australian Synchrotron. The detector was calibrated using silver behenate standards on the Dectris PILATUS3 X 2 M. The intensity was normalised via an integrated diode beamstop. The X-ray beam energy was fixed at 11.001 keV. Using the in-house Scatterbrain programme, the obtained data were condensed to a single dimension. After fitting the SAXS data with the McSAS software using a basic Monte Carlo simulation, the size distribution profile was extracted[57]. A spherical model was fitted to the data.

## Solid-state NMR

Solid-state NMR measurements were performed using a 1.6 mm HXY FastMAS probe on a 600 MHz Varian NMR system, with a sample spinning frequency set at 20 kHz. For the $^{133}Cs$ MAS NMR spectra, a 1 μs excitation pulse, 1200 scans, and a repetition delay of 60 s were utilized, while the $^{1}H$ MAS NMR spectra were obtained using a 1.5 μs excitation pulse, 8 scans, and a repetition delay of 5 s.

In the $^{1}H$-$^{13}C$ CPMAS experiments, a cross-polarization (CP) block of 4 ms, and signal acquisition with high-power XiX proton decoupling were used, with a repetition delay of 1 s and a total of 6000 scans collected. The length of the CP block in the $^{1}H$-$^{133}Cs$ CPMAS experiment was 4 ms, with a repetition delay of 0.6 s and 120,000 scans collected. The Larmor frequencies for $^{1}H$, $^{13}C$, and $^{133}Cs$ were 599.36 MHz, 150.71 MHz, and 78.61 MHz, respectively. The $^{1}H$ and $^{13}C$ spectral axes were referenced relative to the signals of tetramethylsilane, and the $^{133}Cs$ axis relative to the signal of caesium nitrate.

## Computational details

Standard computational chemistry calculations were carried out using the Gaussian 16[58] and AMS-DFTB[59]. For the theoretical study of infrared spectroscopy, we use a model system of a tetrahedral complex between a $Zn^{2+}$ cation surrounded by imidazole and iodide anions. Geometries and vibrational frequencies of these molecular models were obtained with the B3LYP method[60] in conjunction with the $6-31+G(d,p)$[61] basis set. Modelling of bulk $PbCsI_3$ and the MOF materials, as well as the surface-cluster systems, were carried out using the XTB1 method[62]. For bulk materials, the geometries were taken from the crystal structures. For the cluster systems, we used constrained optimization with the $PbCsI_3$ component being fixed at crystal-structure geometries.

## Data availability

All data sets generated during the current study are encompassed within the published article or the accompanying Supplementary Information. Source data are provided with this paper.

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

## Acknowledgements

The authors acknowledge the funding support from the Australian Research Council (ARC) (FT210100589; DP230101901; DE230100173 and FL190100139). We gratefully acknowledge research funding from the Hyogo Science and Technology Association, Japan (Project 4019), and FOCUS Establishing Supercomputing Center of Excellence, Japan. Computational resources were provided by the RIKEN Information Systems Division, Japan (Q22266). This research was supported by an AINSE Ltd. Postgraduate Research Award (PGRA) and Early Career Researcher Grant (ECRG). X.L. acknowledge the support by an Australian Government Research Training Program (RTP) Scholarship. The authors acknowledge the facilities, and the scientific and technical assistance, of the Australian Microscopy & Microanalysis Research Facility at the Centre for Microscopy and Microanalysis, The University of Queensland. This work used the Queensland node of the NCRIS-enabled Australian National Fabrication Facility (ANFF). This research was undertaken on the THz and SAXS beamline at the Australian Synchrotron, part of ANSTO. SMC acknowledges support from the UK Engineering and Physical Sciences Research Council (EPSRC, EP/V044907/1). This project has been funded in part by the European Union's Horizon 2020 research and innovation programme under Grant Agreements 823717 (ESTEEM3). HE-XRD and XAS measurements were performed at the 11-ID-B and 10-ID-B beamlines of the Advanced Photon Source, a user facility operated for the DOE Office of Science by Argonne National Laboratory under Contract No. DE-AC02–06CH11357. Operations at 10-ID-B are further supported by the Materials Research Collaborative Access Team and its member institutions. We would like to thank Drs. Olaf Borkiewicz and Joshua Wright for assistance at 11-ID-B and 10-ID-B respectively. The authors acknowledge DESY (Hamburg, Germany), a member of the Helmholtz Association (HGF) for the provision of PETRAIII. The authors would like to thank Dr Fernando Saldana for assistance with in-situ diffraction measurements performed at the P21 beamline under proposal I-20230942. A.K. and G.M. acknowledge the financial support from the Slovenian Research Agency (research core funding no. P1-0021 and research project J1-50020). We dedicate this paper to the cherished memory of our esteemed colleague, Gregor Mali, who passed away during the reviewing process prior to the publication of this work, and we deeply appreciate his invaluable contributions to our research efforts.

## Author contributions

Conceptualization: X.L. and J.H. data curation and formal analysis: X.L., J.H., A.K., Y.Y., A.S., M.G., I.M., B.C., D.A., P.C., X.W., J.A.S., Q.S., N.M.B., L.H.G.T. and S.M.C.; experimental design, validation, and investigation: X.L., J.H., W.H., J.L., H.G.H., R.L., A.K.C. and A.S.; project administration, funding acquisition, and resources: D.A., G.M., S.M.C., V.C., L.W. and J.H.; original draft: X.L. and J.H.; review and editing: all authors. All authors have approved the manuscript to publish.

## Competing interests

The authors declare no competing interests.
