## [Peer Review File · Nature Communications]

Interfacial alloying between lead halide perovskite crystals and hybrid glassesREVIEWER COMMENTS

Reviewer #1 (Remarks to the Author):

The authors studied the phase composition and interfacial alloying of $(\text{CsPbI}_3)_x(\text{a}_g\text{ZIF-62})_y$ hybrids with respect to the loading proportion of CsPbI_3 and sintering temperature. They achieved an optimal radiative efficiency upon $(\text{CsPbI}_3)_{0.05}(\text{a}_g\text{ZIF-62})_{0.95}$ sintered at 275 °C (below the T_g of $\text{a}_g\text{ZIF-62}$), highlighting that suppresses the formation of non-perovskite $\delta\text{-CsPbI}_3$ and reduces the defective interfacial alloying structures are key to maximize the radiative efficiency of hybrids. They used STEM-CL and synchrotron techniques to identify the luminescence and bonding features of nano-/ fine- structures. They demonstrated the diffusion of the iodine into the glassy phases was significant, accompanied by the preservation of Pb-I octahedron structure within the diffused alloying layer.

This work is interesting, which explores the underlying interfaces of perovskite-glass hybrids (Science, 2021, 374, 621). The rules may be of value to research community who are working on perovskite hybrid luminophores. Some concerns need to be addressed prior to be considered for acceptance by Nature Communications.

1. The PLQY plotted versus sintering temperature and CsPbI_3 loading (Fig. S7) shall be more preferred to alter Fig. 2a, since the PL intensity fails to provide physical implications.

2. The high-energy PL tails are observed for $(\text{CsPbI}_3)_{0.40}(\text{a}_g\text{ZIF-62})_{0.60}$ at lower sintering temperatures (Fig. 2g). Have the authors ever considered the spectral contribution of wide-bandgap $\delta\text{-CsPbI}_3$ beyond interfacial disorders or trap states? Since the $\delta\text{-CsPbI}_3$ previously interpreted the PL broadening of polymorphic CsPbI_3 film (Nat. Photonics, 2020, 15, 238).

3. Somewhat controversial is that the proportion evolution of $\gamma\text{-CsPbI}_3$ retrieved from the Rietveld refinement of the XRD profiles (Fig. 1d) is inconsistent with the results of X-ray pair distribution functions (Fig. 5c, Figs. S33 and S34). Which one is more reliable?

4. Why are the CL-STEM images of $(\text{CsPbI}_3)_{0.10}(\text{a}_g\text{ZIF-62})_{0.90}$ sintered at 350 °C (Fig. S10) the same as Fig. 2?

5. What is the exact PLQY of $(\text{CsPbI}_3)_{0.25}(\text{a}_g\text{ZIF-62})_{0.75}$ and $(\text{CsPbI}_3)_{0.40}(\text{a}_g\text{ZIF-62})_{0.60}$ sintered at 350 °C? Fig. S7 shows that the PLQY of $(\text{CsPbI}_3)_{0.40}(\text{a}_g\text{ZIF-62})_{0.60}$ is seemingly higher than that of $(\text{CsPbI}_3)_{0.25}(\text{a}_g\text{ZIF-62})_{0.75}$, while the EXAFS signals present that $(\text{CsPbI}_3)_{0.40}(\text{a}_g\text{ZIF-62})_{0.60}$ features more defective alloying structures (Fig. 4c-h and Fig. S29).

6. How to differs the interface from the bulk in PDFs regarding its characters for average structure identification?

Reviewer #2 (Remarks to the Author):

The interesting work done by Li et al. demonstrates an answer to address the long-standing ambiguity between surface or bulk properties as the determination of device performance based on CsPbI₃ hybrid glass alloy. They also found the presence of a diffusion alloying as beneficial factor for surface trap passivation. This work is well-organized and results are interesting. I have some comments as shown below:

1. Since it remains a long-standing ambiguity for the origination of device performance, more detailed discussions should be provided in the introduction section. This is critical for readers better understand how surface traps or bulk traps influence device performance and why it has not reached a common conclusion. Meanwhile, it might be too arbitrary to reach a conclusion “optoelectronic performance of composite perovskite was predominately determined by the interface, instead of the bulk particle crystallinity” because only one perovskite composite case is presented in this manuscript.
2. I am curious about the formation of (CsPbI₃)_{0.01}(agZIF-62)_{0.99} alloy. Is CsPbI₃ nanoparticle embedded in agZIF-62 matrix to form core-shell structure like previously reported perovskite/mesoporous silicon composite (Adv. Funct. Mater. 2023, 33, 2210765., Angew. Chem. Int. Ed.2020,59, 23100)? A schematic diagram showing structure is helpful for better understanding.
3. What type of defects for the perovskite surface in the composite? Evidences should be given.
4. In the section of “Stable optical performance”, (CsPbI₃)_{0.05}(AgZIF-62)_{0.95} exhibits a maximum PLQY of 81.3% under sintering at 275°C. The sample has good thermal stability while I am curious about device stabilities under UV stress or in water. Because these are also important for the application in optoelectronic devices. Meanwhile, if environmental conditions change, which should be fatal factor for the materials?
5. Some minor suggestions: In FigureS6 of supporting information, the explanation of the FigureS6 should be consistent with the label of the figure, which should both be a, b, not (A), (B). On page 6 of the text, it is mentioned that samples for STEM are prepared by the focused ion beam method, please add the specific operation of the FBI method. Please adjust the layout of Figure 5 so that it fits the page well.
6. In the conclusions, the authors look ahead to the future about how the interfacial engineering of this composite material will inspire the material properties for practical applications such as solar cells, LEDs, etc. It is also important for the material to be used in practical applications. Please give some examples of how the application of this material can be prospected.

Reviewer #3 (Remarks to the Author):

The manuscript, authored by Li and colleagues, presents a study on the fabrication of an array of CsPbI₃ crystal and hybrid glass composites through sintering. The authors further investigate the inherent relationship between the structure and properties of these materials. The research aims to provide valuable insights into the development of high-performance perovskite devices, with a specific focus on functional hetero-interfaces. However, the current lack of sufficient data and explanation hinders the manuscript's suitability for publication at this stage. Additionally, the work lacks the necessary level of

significance and novelty to meet the standards of a journal like Nature Communications. Please find below some additional comments and suggestions.

1. Firstly, a study on the physicochemical properties between CsPbI₃ and ZIF-62 through sintering has been previously reported by Hou et al. in Science (374, 621–625, 2021). However, the research content discussed in this manuscript lacks significant innovation, despite the comprehensive exploration of the corresponding structure, phase evolution, and changes in PL spectra.
2. Furthermore, the critical conclusion that the interface plays a crucial role in determining the crystal phase, optoelectronic quality, and stability lacks essential interface structure characterization data. There are deficiencies in the key evidence supporting the interpretation of the diffusion "alloying" layer.
3. To provide a better understanding of the phase evolution of the samples, it is necessary to include temperature-resolved, high-resolution in situ XRD characterization of (CsPbI₃)_x(agZIF-62)_y in the manuscript.
4. The magnification of the HAADF image in Figure 2 is too small. Additionally, it is worth noting that several differences reflected in the image contrast do not exhibit complete spatial correspondence with the PL intensity. Further explanation and additional characterization at higher magnifications should be provided.
5. The ordinate value is missing in Figure 2g. Furthermore, we strongly recommend supplementing the PL spectra of the (CsPbI₃)_{0.25}(agZIF-62)_{0.75} composite.
6. Moreover, it is crucial to include high-resolution TEM imaging and corresponding EDS mapping to ensure the atomic structure of the perovskite phase. Specifically, the boundary structure between the perovskite and amorphous phases should be further clarified.
7. In Figure 4c and f, please add the chemical bonds corresponding to different peak positions. Additionally, the changes in radial distance at different sintering temperatures should be quantified and summarized in Figure 4.

Reviewer #4 (Remarks to the Author):

The work by Li et al investigates the interface of CsPbI₃ and ZIF-62 after sintering. The perovskite is a commonly studied material for optoelectronic applications. Pb-based perovskites are problematic due to the toxic element Pb which is regulated in use by EU. Recently, perovskite-MOF composites were identified as improving the properties of the luminescent material.

The study is extremely thorough in analyzing all the diffusion interfaces forming in the process with respect to sintering temperature and the authors correlate this with PL properties. They use XRD, SAXS, DSC, THz spectroscopy, PDF and 133Cs solid state NMR spectroscopy. All methods provide a quite comprehensive picture on the properties of the composites. STEM analysis identifies the interlayer and EELS gives information on composition. In conclusion the authors point out the importance of the interface for PL properties.

Overall this is a quite thorough study with many contributors. The PL analysis is performed on a relative scale, quantum yields are not reported.

A conceptual question is whether MOF-Perovskite composites are promising for EL or PV applications as

the electrically isolating MOF-matrix hampers the charge carrier injection. In this context PL measurements only provide a quite limited information about potential device performance and probably the perspective of such materials is a bit overemphasized.

From a fundamental point of view the study is insightful and worth publication.

RESPONSE TO REVIEWERS' COMMENTS

We would like to express our gratitude to the four independent reviewers, who spent their valuable time reading through this manuscript to offer comments/suggestions to improve the quality of our work. We have reproduced all reviewer comments below in text boxes, with the corresponding point-by-point responses following each comment with the changes being highlighted with blue colour.

Response to Reviewer #1:

Overall comments: The authors studied the phase composition and interfacial alloying of $(\text{CsPbI}_3)_x(\text{a}_g\text{ZIF-62})_y$ hybrids with respect to the loading proportion of CsPbI_3 and sintering temperature. They achieved an optimal radiative efficiency upon $(\text{CsPbI}_3)_{0.05}(\text{a}_g\text{ZIF-62})_{0.95}$ sintered at 275 °C (below the T_g of $\text{a}_g\text{ZIF-62}$), highlighting that suppressing the formation of non-perovskite $\delta\text{-CsPbI}_3$ and reducing the defective interfacial alloying structures are key to maximize the radiative efficiency of hybrids. They used STEM-CL and synchrotron techniques to identify the luminescence and bonding features of nano-/fine-structures. They demonstrated the diffusion of the iodine into the glassy phases was significant, accompanied by the preservation of Pb-I octahedron structure within the diffused alloying layer. This work is interesting, which explores the underlying interfaces of perovskite-glass hybrids (Science, 2021, 374, 621). The rules may be of value to research community who are working on perovskite hybrid luminophores. Some concerns need to be addressed prior to be considered for acceptance by Nature Communications.

We would like to thank the referee for her/his positive comments on the significance and the novelty of this work, as well as the constructive suggestions on improving its quality. Our responses to the specific comments are made as follows.

Comments #1: The PLQY plotted versus sintering temperature and CsPbI_3 loading (Fig. S7) shall be more preferred to alter Fig. 2a, since the PL intensity fails to provide physical implications.

We appreciate this comment and agree that the PLQY figure has significance physical implications. As such, we have now placed it in the main manuscript (Figure 2a) and subsequently moved the PL intensity contour plot to Supporting Information as Figure S8.

Comments #2: The high-energy PL tails are observed for $(\text{CsPbI}_3)_{0.40}(\text{a}_g\text{ZIF-62})_{0.60}$ at lower sintering temperatures (Fig. 2g). Have the authors ever considered the spectral contribution of wide-bandgap $\delta\text{-CsPbI}_3$ beyond interfacial disorders or trap states? Since the $\delta\text{-CsPbI}_3$ previously interpreted the PL broadening of polymorphic CsPbI_3 film (Nat. Photonics, 2020, 15, 238).

We appreciate this insightful comment. The high-energy PL tails of $(\text{CsPbI}_3)_{0.40}(\text{a}_g\text{ZIF-62})_{0.60}$ composites were typically positioned between 2.0 and 2.25 eV (Figure 2g). Wide-bandgap $\delta\text{-CsPbI}_3$ has the potential to broaden the PL emission between 2.06-3.10 eV¹. However, we would like to acknowledge that a more likely explanation is that these tails originate from the MOF glass (the PL spectrum of pure $\text{a}_g\text{ZIF-62}$ PL is provided in SI as Figure S13e), which exhibits

a broader emission over 2.0 eV to 3.3 eV. This finding is supported by Figures S13a-d, which demonstrate that with increased CsPbI₃ loading, the relative contribution from the high-energy PL tail decreased. This phenomenon can be attributed to the fabrication of δ -CsPbI₃ perovskite via a mechanochemical process in the first place, where the presence of a high density of trap states would lead to very low PL emission from the δ -CsPbI₃ perovskite phase before sintering.

Figure S13. PL spectra shown in log scale for (a) (CsPbI₃)_{0.05}(a_gZIF-62)_{0.95} composite, (b) (CsPbI₃)_{0.25}(a_gZIF-62)_{0.75} composite, (c) (CsPbI₃)_{0.50}(a_gZIF-62)_{0.50} composite, (d) (CsPbI₃)_{0.55}(a_gZIF-62)_{0.45} composite, and (e) pure a_gZIF-62.

Comments #3: Somewhat controversial is that the proportion evolution of γ -CsPbI₃ retrieved from the Rietveld refinement of the XRD profiles (Fig. 1d) is inconsistent with the results of X-ray pair distribution functions (Fig. 5c, Figs. S33 and S34). Which one is more reliable?

We would like to acknowledge the thoroughness of the referee. These two techniques measure different structural aspects. The Rietveld refinement and PDF datasets will produce identical fitting parameters only when the long-range, average structure modelled in Rietveld matches the shorter range, local structure seen in the PDF. While this is true for perfect single crystals, materials such as CsPbI₃ composites clearly possess both a crystalline fraction, and a more disordered fraction. The Rietveld probes only the fraction with long range order, while the PDF captures both crystalline and partially amorphous fractions. From a practical perspective, the Rietveld refinement subtracts broad, diffuse scattering as part of the polynomial background underneath the peaks, while these features are fully included in the PDF calculation. This is why the PDF approach is often called “total scattering”.

The two datasets were also collected on different instruments, but we do not believe this is the origin of the discrepancy. The PDF data (wide angular range, lower angular resolution) was measured with a high energy synchrotron beam in transmission mode while the Rietveld data (lower angular range, higher angular resolution) originates from a laboratory source in Bragg-Brentano reflection geometry. We have previously collected complementary synchrotron Rietveld data on several other pelletized CsPbI₃ samples and observe negligible compositional variation near the surface vs the core of the pellet, and across the length of the pellet at the scale of the beam (tens of μm). While we have noticed X-ray beam damage artifacts with previous *in situ* synchrotron experiments, the short single exposures used here to collect the PDF are well below this threshold.

Comments #4: Why are the CL-STEM images of (CsPbI₃)_{0.10}(a_gZIF-62)_{0.90} sintered at 350 °C (Fig. S10) the same as Fig. 2?

We thank the referee for identifying this typo in the Figure S10 caption. The sample name in the SI Figure S10 should be (CsPbI₃)_{0.40}(a_gZIF-62)_{0.60} and we have updated this in the revised manuscript (as Figure S11). The purpose of including the SI figure is to provide a larger view of the fitting results, also the wavelength parameter not included in Figure 2.

Comments #5: What is the exact PLQY of (CsPbI₃)_{0.25}(a_gZIF-62)_{0.75} and (CsPbI₃)_{0.40}(a_gZIF-62)_{0.60} sintered at 350 °C? Fig. S7 shows that the PLQY of (CsPbI₃)_{0.40}(a_gZIF-62)_{0.60} is seemingly higher than that of (CsPbI₃)_{0.25}(a_gZIF-62)_{0.75}, while the EXAFS signals present that (CsPbI₃)_{0.40}(a_gZIF-62)_{0.60} features more defective alloying structures (Fig. 4c-h and Fig. S29).

For samples sintered at 350 °C, the PLQY of (CsPbI₃)_{0.25}(a_gZIF-62)_{0.75} composite was slightly higher than that of (CsPbI₃)_{0.40}(a_gZIF-62)_{0.60} composites (2.17% vs 1.23%). This result is consistent with our EXAFS analysis, which was performed on these two samples for the same Pb L₃ and Zn K edges. From the EXAFS experiments it was found that the intensity of signals arising from ideal crystal coordination in the (CsPbI₃)_{0.25}(a_gZIF-62)_{0.75} composite is higher than those retrieved from (CsPbI₃)_{0.40}(a_gZIF-62)_{0.60}, indicating that (CsPbI₃)_{0.40}(a_gZIF-62)_{0.60} contains more defective coordination sites around these target atoms.

It follows that these two analyses are in good agreement. The referee touches on an important point here, however, and we have subsequently brought more clarity to these data and related discussion in the revised manuscript.

“Furthermore, all of the peak intensities in the $(\text{CsPbI}_3)_{0.25}(\text{a}_g\text{ZIF-62})_{0.75}$ EXAFS plot are higher than those of $(\text{CsPbI}_3)_{0.40}(\text{a}_g\text{ZIF-62})_{0.60}$, indicating that the $(\text{CsPbI}_3)_{0.40}(\text{a}_g\text{ZIF-62})_{0.60}$ contains more defective site, which is consistent with their corresponding PLQY results (2.17% vs 1.23%).”

Comments #6: How to differs the interface from the bulk in PDFs regarding its characters for average structure identification?

The PDF signal contains the weighted average local environment of all scattering atoms, and therefore detects both ordered bulk phases as well as disordered interfacial regions. In some exceptionally defined media, it is possible to extract the local structure of the interface from the bulk signal and explicitly model it².

However, our CsPbI_3 composites have much too complex and poorly defined structure to accommodate a precise, double-differential PDF analysis. To resolve the disordered phase from the more crystalline fraction, either a boxcar analysis or long-range differential approach can be applied. In our case the latter is more powerful and involves fitting the PDF over a broader range than normally used, up to 50 Å, instead of just the first couple coordination shells. The resulting difference pattern, normally considered the “error” in a refinement, contains the correlation lengths which are relatively over- or under-represented in the disordered phase vs the ordered one.

Of course, this analysis in isolation cannot identify where the disordered phases exist, only their structure, and therefore needs to be supported by the numerous other analytical techniques used in this work to arrive at an assignment of disordered interfacial regions. We realise such details were absent in the original submission and have taken steps to remedy this in our revised work.

“Atomic pair distribution functions (PDFs), a technique which unavoidably measures the average structure of the whole system. In this work, two physical features aid us to differ the interface from the bulk; (i) a relatively high surface to volume ratio in the target crystal beneficially weights interface-related signals, and (ii) there exists clear contrast between the short-range fine structure existing at the alloying layer compared to the long-range ordering in the bulk, with the alloying layer being heavily disordered, deviating from the prominent periodic fine structure found in the bulk. This analysis identified the interfacial diffusion layers mainly contained Pb-I, Cs-I and Pb-N atomic pairs with only short-range orders. *Ex-situ* synchrotron X-ray total scattering measurements were performed on a series of composites sintered at 350 °C. The PDF, obtained by Fourier transform, showed the short-to-long range fine structure of the composites (Figure 5c). Rietveld analysis of the composite $(\text{CsPbI}_3)_{0.50}(\text{a}_g\text{ZIF-62})_{0.50}$ was performed with desolvated ZIF-62, γ - and δ - CsPbI_3 reference structures. We found that ZIF structure was only maintained in the short-range order. Good fitting was achieved in the medium and long-range (R_{wp} of 11 % in the range of 15-50 Å), with

13.7 wt% of γ -CsPbI₃ and 30.1 wt% of δ -CsPbI₃ being identified. The difference pattern in the short-range suggested the structural distortion originated from the surface of the crystalline phases, with the main feature being assigned to Pb-I (*ca.* 3.2 Å). This is an indication of presence of discrete, rigid and slightly distorted Pb-I octahedra within the diffusion alloying layer. These difference features are highly composition- and sintering temperature dependent, showing more contribution from these amorphous perovskite components at higher sintering temperatures, or from composites with larger expected interfaces (Figure S36-37). These observations further confirm the amorphous structures are originated from the interface, instead of the amorphous regions of the bulk perovskites.”

Response to Reviewer #2:

Overall comments: The interesting work done by Li et al. demonstrates an answer to address the long-standing ambiguity between surface or bulk properties as the determination of device performance based on CsPbI₃ hybrid glass alloy. They also found the presence of a diffusion alloying as beneficial factor for surface trap passivation. This work is well-organized, and results are interesting. I have some comments as shown below:

We are pleased that the reviewer recognises the importance of this work. We have addressed the reviewer's valuable comments point-by-point below and revised our manuscript accordingly.

Comments #1: Since it remains a long-standing ambiguity for the origination of device performance, more detailed discussions should be provided in the introduction section. This is critical for readers better understand how surface traps or bulk traps influence device performance and why it has not reached a common conclusion. Meanwhile, it might be too arbitrary to reach a conclusion "optoelectronic performance of composite perovskite was predominately determined by the interface, instead of the bulk particle crystallinity" because only one perovskite composite case is presented in this manuscript.

We thank the reviewer for this insightful observation. The introductory discussion of how surface traps or bulk traps affect device performance has been updated as follows:

"One common strategy for addressing these challenges associated with pure perovskite compounds is to composite them with secondary components^{3,4}. Within this context, organic ligands, polymers, inorganic zeolites and glasses, and recently metal-organic frameworks (MOFs) have all been studied, with each type having its own advantages, and simultaneously disadvantages⁵⁻⁹. Understanding the behaviour and function of the interface inside the composites is critical. Such knowledge will further aid in the development of more efficient and stable devices: perovskites are typically sandwiched between electron and hole transport layers in LEDs and solar panel systems, and the properties of these interfaces can be critical to device performance¹⁰. Surface traps have long been a source of confusion and usually detrimental to the performance and stability of perovskites¹¹, even though they are typically considered as defect-tolerant materials¹². On the other hand, chemical disorder can also capture diffusing carriers over micrometre-length scales, resulting in radiative recombination, outcompeting the capture of carriers in more electronically disordered and trap-rich regions¹³. However, studying both positive and negative effects from the interface remains a difficult task since most high-performing composites are not sufficiently stable to keep their original properties/functions against prolonged exposure to handling and inspection^{14,15}. Other composites may lack a distinct chemical and/or physical contrast between two phases, reducing the capacity to extract important interfacial information¹⁶."

As APbI₃ family share a common optoelectronic backbone of Pb-I orbitals, the conclusion section has been altered to be more precise.

"This work also unveiled the complex performance – property relationship within the CsPbI₃ glass composites. We identified that the optoelectronic performance of CsPbI₃ composites

was predominately determined by the interface, instead of the bulk particle crystallinity. The interfacial alloying favoured the formation of active CsPbI_3 phases, stabilised the optoelectronic performance and, more importantly, effectively passivated the trap states. Nevertheless, it can also be detrimental to light emission: for all composites with varying CsPbI_3 loadings, the maximum PL intensity was obtained by sintering at a temperature at the corresponding T_g . Above the T_g , glass enters a viscous flowing state, leading to faster diffusion and generate non-stoichiometric perovskite near the interface. Though the coarsening can promote the overall crystallinity, the diffusion of Pb-I away from CsPbI_3 can lead to non-radiative recombination and quenching of PL. The principle can be expanded to other ABX perovskite systems because they share a common optoelectronic backbone of metal-halide orbitals.

In addition, we have expanded the conclusion part to demonstrate the boarder implications of this work:

“In particular, the notable stability under UV irradiation and relatively high temperature allows these composites to be further processed by an ultrafast laser. The composites generated a localised liquid phase as a result of the intense thermal accumulation, which can be used to fabricate optical products by three-dimensional printing. The whole process can be adjusted by optimising the pulse duration, repetition rate, and pulse energy as well as ultrafast-laser irradiation time. Furthermore, some advanced fabrication methods like chemical vapor deposition and atomic layer deposition, are promising to reduce the thickness of MOF glass to nano-meter size, making these composites more compatible with planar device components. It is important to keep in mind that when using these composites in real-world applications, the material's limitations should never be disregarded. For instance, the ZIF glass is prone to acid and powerful chelating agents' chemical attacks and can begin to decompose at a temperature of 600 °C. This indicates that when facing the mentioned problem in the design of practical devices, composites may require further encapsulation.”

Comments #2: I am curious about the formation of $(\text{CsPbI}_3)_{0.01}(\text{a}_g\text{ZIF-62})_{0.99}$ alloy. Is CsPbI_3 nanoparticle embedded in $\text{a}_g\text{ZIF-62}$ matrix to form core-shell structure like previously reported perovskite/mesoporous silicon composite (Adv. Funct. Mater. 2023, 33, 2210765., Angew. Chem. Int. Ed.2020,59, 23100)? A schematic diagram showing structure is helpful for better understanding.

For the composite containing only 1 wt % of the perovskites, the structure is expected to be a mixed matrix type, where the MOF glass acts as the substrate, and the perovskite crystal serves as the dispersing phase. Due to the MOF glass being relatively continuous in the composite, it does not form the conventional particle "core-shell" configuration. Instead, the perovskite crystals disperse within the MOF glass matrix.

To better showcase the structure of the composite, an updated scheme has been added to Figure 1a.

Figure 1. **Fabrication of $(\text{CsPbI}_3)_x(\text{a}_g\text{ZIF-62})_y$ composites.** (a) Schematic diagram of the CsPbI_3 phase transition and evolution of the interfacial atomic structures during sintering. Arrows indicate the progress of sintering. (b) Schematic diagram of the DFT calculation for composites with different CsPbI_3 crystal phases. (c) *Ex-situ* XRD pattern of $(\text{CsPbI}_3)_x(\text{a}_g\text{ZIF-62})_y$ composites sintered at 350 °C. X-ray λ : 1.5406 Å. (d) γ -phase CsPbI_3 proportion within of different $(\text{CsPbI}_3)_x(\text{a}_g\text{ZIF-62})_y$ composites, as retrieved from Rietveld refinement of the XRD profiles.

Comments #3: What type of defects for the perovskite surface in the composite? Evidence should be given.

We appreciate this valuable comment. Given that both pure CsPbI₃ and its composites were fabricated by using a ball milling technique and have a low crystallinity, many types of defects could be expected from each component prior to sintering. The sintering improves the crystallinity of the perovskites, reducing the bulk phase defects. We have further identified that the sintering process (at a temperature corresponding to T_g of the different composites) could be a defect passivation process at the interface, where we can extract significant information from the changes of chemical bonding. For instance, the emerging Zn-I, Pb-Zn and Pb-N, as well as the vanished Pb-O and increased Pb-I bond types. According to first-principles density functional theory (DFT) calculations, the formation of Zn-I can be attributed to Zn ions that take the place of V_{Cs} (Cs vacancy), which has the lowest transition level¹⁷. Moreover, the microscopic mechanism of ZIF melting involves the breaking and reformation of Zn-N bonds, which provide great opportunity for Pb ions to coordinate with imidazolate-termination sites, resulting in new Pb-N bonds. There are two possible pathways forming Pb-Zn atomic pairs: When CsPbI₃ crystallises during sintering, Pb-O bonding was eliminated, resulting in Pb-I becoming more dominant, during which a minor portion of Pb ions interacted with Zn-termination. Another condition is that Zn ion replaces V_I (I vacancy) from the diffused alloying layer¹⁸.

Based on the above discussion, some updates have been included in the main text:

“The local environment of the Zn prior to sintering was identical in the crystalline ZIF-62¹⁹, indicating Zn atoms maintains their tetrahedral configuration within the composites. After sintering, the emerging peak at *ca.* 2.7 Å can be assigned to Zn-I, in good agreement with the THz results, which are assigned to Zn ions taking the place of \$V_{Cs}\$ (Cs vacancy) as its lowest transition level among all kind of defects¹⁷.”

“In addition, the emerging peak at *ca.* 1.4 Å and 2.5 Å can be attributed to Pb-N and Pb-Zn pairs formed at the interfaces. The formation of new atom pairs was supported by the X-ray adsorption near-edge structure (XANES) of the Pb L₃-edge spectra (Figure S28-29), which exhibit a shift toward lower energy after sintering, due to the higher electronegativity of N and Pb over Zn. The microscopic mechanism of ZIF melting involves the breaking and reformation of Zn-N bonds, which permits Pb ions to coordinate with imidazolate-termination sites, resulting in new Pb-N bonds. Meanwhile, while a minor portion of Pb ions interacted with Zn-termination or Zn ion replaces \$V_I\$ (I vacancy) from the diffused alloying layer forming Pb-Zn¹⁸.”

Comments #4: In the section of “Stable optical performance”, (CsPbI₃)_{0.05}(a_gZIF-62)_{0.95} exhibits a maximum PLQY of 81.3% under sintering at 275°C. The sample has good thermal stability while I am curious about device stabilities under UV stress or in water. Because these are also important for the application in optoelectronic devices. Meanwhile, if environmental conditions change, which should be fatal factor for the materials?

We agree that high stability is essential for these materials to be used in devices. Prior to this work, a thorough stability assessment was conducted on these materials ((CsPbI₃)_{0.25}(a_gZIF-

62)_{0.75}), where we observed significant stability against 10,000 hours of submersion in water, 650 days of storage at room temperature, mild heating, and continuous laser excitation (~57 mW/cm²) for more than 5000 s²⁰. Higher stability would be expected in the current case, given that the smaller amount of perovskite can be better wrapped and protected by the glass substrates. The central goal of this paper is to address the fundamental questions related to the interface within the composites, and the stability data therefore were not reproduced nor studied in depth. We recognise now that the issue of stability was somewhat brushed over, and we have now remedied this in the revised work by expanding our discussion of this important feature in conclusion part.

“In particular, the notable stability under UV irradiation and relatively high temperature allows these composites to be further processed by an ultrafast laser.”

Regarding the second comment regarding the fatal vulnerability of the system, we find the main weakness of the composite originates from the glass phase, i.e., the thermal and chemical stability of the protective substrate. According to our previous study, the ZIF glass can start to decompose at a temperature of 600 °C and it is susceptible to the chemical attack by acid and strong chelating agents. This could potentially be addressed by additional encapsulation into real devices. Some suggestions have been presented in the conclusion section.

“It is important to keep in mind that when using these composites in real-world applications, the material's limitations should never be disregarded. For instance, the ZIF glass is prone to chemical attack by acid and strong chelating agents' chemical attacks and can begin to decompose at a temperature of 600 °C. This indicates that when facing the mentioned problem in the design of practical devices, composites may require further encapsulation.”

Comments #5: Some minor suggestions: In FigureS6 of supporting information, the explanation of the FigureS6 should be consistent with the label of the figure, which should both be a, b, not (A), (B). On page 6 of the text, it is mentioned that samples for STEM are prepared by the focused ion beam method, please add the specific operation of the FBI method. Please adjust the layout of Figure 5 so that it fits the page well.

We value the referee's opinion. All figures in the Manuscript and SI have been thoroughly examined, and the labels and figures in Figures S7, S9, S11, S12, S28, S29, S32, and S33 have been unified. Besides, to make Figure 5 fits the page better, the layout has been adjusted as follows.

Figure 5. Chemistry and structure of the alloying layer at the interface. (a) ADF-STEM image of $(\text{CsPbI}_3)_{0.1}(\text{agZIF-62})_{0.9}$ composites sintered at 350 °C. (b) EELS core loss spectra acquired from the marked points shown in TEM image with background subtracted. (c-d) Atomic pair distribution functions for $(\text{CsPbI}_3)_{0.50}(\text{agZIF-62})_{0.50}$ composites sintered at 350 °C. Pair distribution function $D(r)$ calculated *via* Fourier transform of the X-ray total scattering function. Panel (d) is the enlarged fitted data in the r range of 1.5-12 Å.

Following the description of SEM, a detailed FIB/SEM procedure was provided in Methods, SI.

“Focused ion beam scanning electron microscopy (FIB-SEM)

An FEI SCIOS focused ion beam scanning electron microscope (FIB/SEM) dual beam system was employed to generate TEM lamella. The area of interest on the sample surface was coated with a 1 μm thick layer of platinum to shield it from ion bombardment damage. Subsequently, applying a Ga ion beam current ranging from 3 nA to 30 nA at an acceleration voltage of 30 kV, a thin slice of the material was removed perpendicularly to the sample surface by milling two trenches on both sides. After being detached from the sample matrix, the thin segment was bonded on a TEM half-grid after being mounted to the EazyLift Micromanipulator. Following post-thinning with a significantly reduced ion beam current (from 0.3 nA down to 30 pA), low voltage polishing at grazing incidence was used to complete the cleaning process.”

Comments #6: In the conclusions, the authors look ahead to the future about how the interfacial engineering of this composite material will inspire the material properties for practical applications such as solar cells, LEDs, etc. It is also important for the material to be used in practical applications. Please give some examples of how the application of this material can be prospected.

We appreciate the referee for this constructive comment. As introduced above, the composites are stable under UV irradiation and high temperature, which can be used for three-dimensional printing. Relevant content has been added to the revised concluding text.

“In particular, the notable stability under UV irradiation and relatively high temperature allows these composites to be further processed by an ultrafast laser. The composites generated a localised liquid phase as a result of the intense thermal accumulation, which can be used to fabricate optical products by three-dimensional printing. The whole process can be adjusted by optimising the pulse duration, repetition rate, and pulse energy as well as ultrafast-laser irradiation time. Furthermore, some advanced fabrication methods like chemical vapor deposition and atomic layer deposition, are promising to reduce the thickness of MOF glass to nano-meter size, making these composites more compatible with planar in device components.”

Response to Reviewer #3:

Overall comments: The manuscript, authored by Li and colleagues, presents a study on the fabrication of an array of CsPbI₃ crystal and hybrid glass composites through sintering. The authors further investigate the inherent relationship between the structure and properties of these materials. The research aims to provide valuable insights into the development of high-performance perovskite devices, with a specific focus on functional hetero-interfaces. However, the current lack of sufficient data and explanation hinders the manuscript's suitability for publication at this stage. Additionally, the work lacks the necessary level of significance and novelty to meet the standards of a journal like Nature Communications. Please find below some additional comments and suggestions.

We thank the reviewer for carefully reading our manuscript. We have addressed the reviewer's valuable comments point-by-point below and revised our manuscript accordingly.

Comments #1: Firstly, a study on the physicochemical properties between CsPbI₃ and ZIF-62 through sintering has been previously reported by Hou et al. in *Science* (374, 621–625, 2021). However, the research content discussed in this manuscript lacks significant innovation, despite the comprehensive exploration of the corresponding structure, phase evolution, and changes in PL spectra.

We appreciate this feedback. As the reviewer mentioned, our previous work published in *Science* places a significant emphasis on the global exploration of perovskites and a_gZIF-62 composites in terms of their fabrication and physicochemical features including stability and optoelectronic performance. Due to the environmental sensitivity of perovskites, studying the interface inside different types of perovskite composites has proven to be particularly challenging but is key to deciphering their practical device behaviours. In this work, we take full advantage of the rich chemical, structural properties, strong contrast and satisfactory stability of these specific composites, using this unique platform to investigate the detailed mechanism and the evolution of the sintering process. Specifically, we studied the important roles of interfacial properties through the composites with different sintering temperatures and components. Multiple techniques combined with corresponding calculations were conducted to explore the structure evolution during the sintering process. This work reveals the fine structure of the interface and identifies the key parameters for optoelectronic performance. As a result, the model system can provide essential knowledge that can be used to drive the development of further perovskite devices and other perovskite composites.

Comments #2: Furthermore, the critical conclusion that the interface plays a crucial role in determining the crystal phase, optoelectronic quality, and stability lacks essential interface structure characterization data. There are deficiencies in the key evidence supporting the interpretation of the diffusion "alloying" layer.

We value the comments from the reviewer. In this work, multiple techniques involved in the evolution of particle size, chemical bonding, and element distribution were used to investigate the diffusion alloying layer. In particular, synchrotron X-ray small angle scattering (SAXS) demonstrated the coarsening of the particle during the sintering process, demonstrating that

the viscous flow rates of the MOF liquid aided the diffusion of the perovskite particles. Further analysis of SAXS data with the Porod's plot revealed that higher sintering temperatures caused a greater positive deviation from the Porod's law, showing the system no longer have a sharp phase boundary with clear contrast of electron density along with sintering, indicating the evolution of a diffusion layer.

By using extended X-ray absorption fine structure (EXAFS) and wavelet transforms (WTs), as well as in situ temperature-resolved synchrotron terahertz (THz) FarIR vibrational spectroscopy, new chemical bondings/environments were discovered. These included gradually intensified Zn-I, as well as Pb-N and Pb-Zn pairs-which can only be formed at the interfaces. The interfacial diffusion layer was further proven by careful refinement of the differential X-ray atomic pair distribution function (PDF) patterns that mainly contained Pb-I, Cs-I and Pb-N atomic pairs, exclusively with only short-range orders.

Further, an asymmetric 'shoulder' of the γ -CsPbI₃ peaks were revealed by solid state NMR (SS NMR), which was related to the γ -CsPbI₃ impacted by the chemical environment at the interface, directly in contact with glassy matrix, forming gradient interdiffusion and modified bonding at the interface. Additionally, ¹H-¹³³Cs cross-polarization magic angle spinning (CPMAS) NMR showed a non-negligible cross-polarization transfer between the protons of a_gZIF-62 and the "shoulder-peak" of γ -CsPbI₃ at the interface, indicating a close proximity between the imidazole ligands and Cs atoms.

Moreover, an excess quantity of I outside the CsPbI₃ (350°C) is microscopically located using a line scan across the interface by TEM electron energy loss spectroscopy (EELS), indicating the gradual diffusion of heavier I atoms into the glassy phase at high temperature. This observation aligns with the micro-fluctuation in electron density as demonstrated in SAXS.

More importantly, we have also identified that the PL intensity is not directly correlated to the crystallinity of the perovskite phase, where higher sintering temperature leads to better crystal quality but quickly quench the PL intensity without significant weight loss (additional TGA results in Figure S16b). This result is counter-intuitive to previous works, which aim for increased crystallinity alone. Collectively, we can reasonably attribute the loss of PL is caused by the interfacial interactions (i.e., the formation of the bonding and subsequent alloying layer during sintering), providing new grounds for important device optimisation for researchers working on more applied topics.

We have update Figure S16 as follows:

Figure S16. DSC and TGA results upon sintering to 350 °C. Data was collected under constant flowing nitrogen (20 mL/min). (a) DSC results for melt-quenched a_g ZIF-62 and (b) TGA results for $(\text{CsPbI}_3)(a_g\text{ZIF-62})(40/60)$. The temperature ramp rate was 20 °C/min.

Comments #3: To provide a better understanding of the phase evolution of the samples, it is necessary to include temperature-resolved, high-resolution *in situ* XRD characterization of $(\text{CsPbI}_3)_x(a_g\text{ZIF-62})_y$ in the manuscript.

We have acted on this advice and now include *in situ* high-temperature synchrotron-based powder diffraction experiments on multiple composites to fully elucidate the perovskite crystal phase transition process during the sintering and quenching processes (Figure S5).

For the $(\text{CsPbI}_3)(a_g\text{ZIF-62})(5/95)$ sample, new peaks emerged (from ca. 190 °C) during the sintering process can be assigned to cubic α - CsPbI_3 , and they intensified upon higher temperature sintering. During the cooling ramp, these cubic structure phases remained relatively unchanged above 290 °C, and then the features associated with β - CsPbI_3 ($P4/mbm$) gradually emerged from ca. 270 °C, followed by γ - CsPbI_3 ($Pbnm$) formation from ca. 170 °C. Similar behaviour was observed with $(\text{CsPbI}_3)(a_g\text{ZIF-62})(40/60)$ during the *in-situ* test. Further increase of the CsPbI_3 loading could not effectively preserve the optoelectronically active phase after quenching. These results are consistent with the Rietveld refinement of the XRD profiles summarised in Figure 2d.

The test details were added the SI methods section.

“Synchrotron powder XRD analysis

In situ synchrotron x-ray powder diffraction data was collected at PETRA III beamline P21.1 at the Deutsches Elektronen-Synchrotron (DESY), Hamburg, Germany. For high-resolution XRD, a defocused x-ray beam (101 keV, $\lambda = 0.10173 \text{ \AA}$) was applied, with detector distance of 1950 mm. During the measurement, a steady flow of nitrogen was introduced into a 1.3 mm quartz capillary containing the sample. The sample was heated with Linkam TS1500 at a ramping rate of 20 °C/min to 350 °C and cooled back to room temperature after an isotherm stage for ~100 s at the highest temperature. To keep the phase fractions constant, the diffraction intensities were normalised to the integrated signal at high angles.”

We also add the relevant discussion in the main text:

“*In-situ* XRD was used to track the sintering-induced phase changes in $(\text{CsPbI}_3)_x(\text{a}_g\text{ZIF-62})_{1-x}$ (Figure S4), showing the emergence of active phase CsPbI_3 from 150°C , which was substantially lower than the intrinsic phase transition temperature of pure CsPbI_3 near equilibrium (320°C). Further heating improved the crystallinity and the proportion of perovskite crystal formed. High resolution *in situ* synchrotron powder XRD further proved the phase evolutions are highly dependent on the loading of the perovskites, where the optoelectronically active phase cannot be effectively preserved within composite when the CsPbI_3 loading is higher than 40 % (Figure S5). After sintering, the composite showed a smooth and continuous surface morphology, with clear phase contrast under scanning electron microscope (SEM, Figure S6).”

Figure S5 High resolution *in situ* synchrotron powder XRD for different samples during sintering and quenching. The dominating CsPbI_3 phases are color-coded as: δ (yellow), α (red), β (blue) and γ (grey). (a) $(\text{CsPbI}_3)_x(\text{a}_g\text{ZIF-62})_{1-x}$ (5/95), (b) $(\text{CsPbI}_3)_x(\text{a}_g\text{ZIF-62})_{1-x}$ (40/60) and (d) $(\text{CsPbI}_3)_x(\text{a}_g\text{ZIF-62})_{1-x}$ (55/45).

Comments #4: The magnification of the HAADF image in Figure 2 is too small. Additionally, it is worth noting that several differences reflected in the image contrast do not exhibit complete spatial correspondence with the PL intensity. Further explanation and additional characterization at higher magnifications should be provided.

The field of view of the HAADF image in Figure 2 matches the CL data cube acquisition field of view. The limitation in STEM acquisition is the sampling (area per pixel). The choice of sampling for these experiments was based on the sensitivity of the CsPbI₃ nanocrystals to the electron beam. In short, acquisitions with increased sampling led to emission quenching. Hence our decision to keep this sampling. We tried, but higher sampling in this specific chemical composition does not bring new relevant information in CL experiments.

A large field of view was chosen in order to show the morphology of the sample in a wider area. It was our choice to show that not all objects that appear bright in the HAADF image emit light. This can occur for two reasons. The first one is intrinsic to the material: not all nanocrystals have structure that yields a bright in emission, as previously discussed²⁰. Also, the results presented in this manuscript were acquired from a FIB lamella. In these milled sections, some of the crystal will be exposed at the surfaces of the lamella. These cut or exposed crystals will appear bright in HAADF-STEM, but will most probably not emit light efficiently and quench quickly.

To help clarify these points in the text, we have made the following amendments:

“In this field of view, selected to highlight a wide distribution of particles while still resolving individual grains, the variation of the emission intensity was not directly related to the size of the particles, indicating the quantum confinement was not the main contributor to the strong light emission.”

“It is worth mentioning that to achieve single particle emission profile STEM lamellae samples were prepared by focused ion beam (FIB), which is known to be prone to damaging perovskites and to quenching CL emission. Some nanocrystals visible in the STEM image, particularly those at the surfaces of the lamella, are accordingly not expected to show strong emission. However, the FIB lamellae emitted light at the same wavelength and with the same emission spectral shape as those powder samples prepared by crushing the composite using a mortar and a pestle, confirming the strong stabilisation effects within the composites.”

Comments #5: The ordinate value is missing in Figure 2g. Furthermore, we strongly recommend supplementing the PL spectra of the (CsPbI₃)_{0.25}(a_gZIF-62)_{0.75} composite.

We thank the reviewer for the comment. However, The PL spectra shown in Figure 2g and Figure S13 is in arbitrary unit, so we did not supply the ordinate value in the y-axis. The PL spectra of the (CsPbI₃)_{0.25}(a_gZIF-62)_{0.75} composite is supplied in the Figure S13b.

Comments #6: Moreover, it is crucial to include high-resolution TEM imaging and corresponding EDS mapping to ensure the atomic structure of the perovskite phase. Specifically, the boundary structure between the perovskite and amorphous phases should be further clarified.

This class of crystal-glass composite materials has undergone significant microscopic characterization of the atomic structure of the perovskite phase, including electron diffraction from individual grains as well as electron tomography of the interface reported previously²⁰. The diffracting regions (crystals) are known to exhibit sharp boundaries, meaning the interface structure is disordered, precluding atomic resolution imaging of the material just outside the grain using high resolution TEM or STEM.

We have carefully considered this comment, and we agree that additional characterisation can further corroborate structural details of these particular materials, especially FIB lamellas extracted from the bulk crystal-glass composite.

We have added further high-resolution ADF-STEM data and STEM-EDS mapping of the CsPbI₃ in a_gZIF-62 as prepared in the FIB lamella configuration. We note these confirm the sharp extension of the crystalline lattice to the facets (edges) while also indicating a 10 nm or less further extent of I relative to Cs in EDS as observed in EELS. Notably, EELS offers superior signal collection efficiency (acquiring the vast majority of the inelastic events compared to a few percent of the X-rays emitted recorded in EDS). Outside of very stable samples (under electron beam exposure), EDS data is typically noisier and suffers from loss of spatial resolution due to scattering of electrons through thick samples resulting in X-ray emission from a larger spot than probed in EELS or STEM imaging. We have added further comments in the text to contextualise the additional characterisation data and implications for the boundary structure.

SI Methods:

“Additional ADF-STEM and STEM-based X-ray energy dispersive spectroscopy (STEM-EDS) were acquired using an FEI Titan³ Themis (Thermo Fisher) equipped with a high-brightness ‘X-FEG’ electron source and a four-quadrant (0.7 sr) Super-X EDS detector (Bruker) and operated at 300 kV.”

Main text:

“The atomic structure and spatial variation in elemental composition was further corroborated by ADF-STEM imaging and STEM-based energy dispersive spectroscopy (STEM-EDS) (Figure S34-S35), showing an abrupt termination of the crystals surrounded by a non-crystalline boundary consistent with prior diffraction experiments²⁰ and a similar extent of I outside of the CsPbI₃ crystals.”

Figure S34. (a) Lattice-resolved ADF-STEM image of a FIB lamella showing a CsPbI₃ nanocrystal in a_gZIF-62. (b)-(c) The fast Fourier transform (FFT) of the crystal image was indexed to γ-CsPbI₃ based on the approximately square pattern of spots at ~6 Å (real-space spatial frequency). Annotations in yellow highlight the position of spots and their approximately square symmetry.

Figure S35. STEM-EDS analysis of CsPbI₃ nanocrystals in a_gZIF-62. (a) An overview ADF-STEM micrograph and (b) corresponding line profiles marked in orange. The line profiles show the Cs L α and I L α X-ray intensity at selected grains. The I L α intensity extends further with the Cs L α intensity decaying faster (magenta shaded regions), corroborating the ~5-10 nm further extension of I intensity observed in EELS beyond the detection of Cs in the nanocrystals. (c) EDS maps showing the major constituents of a_gZIF-62 (C, N, Zn) and CsPbI₃.

Comments #7: In Figure 4c and f, please add the chemical bonds corresponding to different peak positions. Additionally, the changes in radial distance at different sintering temperatures should be quantified and summarized in Figure 4.

We would like to thank the referee for this valuable comment. We have updated the Figure 4c and f with marked chemical bonds as follows.

Figure 4. Structure of the interface. (a) ¹³³Cs MAS NMR spectra of (CsPbI₃)_{0.25}(a_gZIF-62)_{0.75} composites sintered at 200, 250, 300, and 350 °C. (b) ¹H-¹³³Cs CPMAS NMR spectra of (CsPbI₃)_{0.25}(a_gZIF-62)_{0.75} sintered at 250 and 350 °C. (c-e) Extended X-ray absorption fine structure (EXAFS) signal and the full-range wavelet transform (WT) representation for the Zn K edge of (CsPbI₃)_{0.40}(a_gZIF-62)_{0.60} composites. (f-h) Extended X-ray absorption fine structure (EXAFS) signal and the full-range wavelet transform (WT) representation for the Pb L₃ edge of (CsPbI₃)_{0.40}(a_gZIF-62)_{0.60} composites. Composites were sintered at 350 °C.

In addition, the changes of radial distance are summarised in SI Table S1 as follows.

Table S1. Results of the extended X-ray absorption fine structure (EXAFS) fitting for $(\text{CsPbI}_3)_{0.40}(\text{a}_g\text{ZIF-62})_{0.60}$ composites before and after 350°C sintering for Zn K and Pb L_3 edge.

Name	Coordination number (CN)	Bond length	Sintering	R-factor
Zn-N	5.07±0.26	1.99±0.004	Before	0.007
Zn-C	6.96±1.35	3.02±0.018	Before	0.007
Pb-O	3.27±0.35	2.46±0.007	Before	0.005
Pb-I	1.69±0.16	3.13±0.011	Before	0.005
Zn-N	5.04±0.31	1.98±0.005	After	0.007
Zn-C	4.59±1.10	2.99±0.020	After	0.007
Pb-I	6.80±1.23	3.12±0.042	After	0.010
Pb-Zn	4.08±0.47	1.80±0.013	After	0.010

Response to Reviewer #4:

Overall comments: The work by Li et al investigates the interface of CsPbI₃ and ZIF-62 after sintering. The perovskite is a commonly studied material for optoelectronic applications. Pb-based perovskites are problematic due to the toxic element Pb which is regulated in use by EU. Recently, perovskite-MOF composites were identified as improving the properties of the luminescent material.

The study is extremely thorough in analysing all the diffusion interfaces forming in the process with respect to sintering temperature and the authors correlate this with PL properties. They use XRD, SAXS, DSC, THz spectroscopy, PDF and 133Cs solid state NMR spectroscopy. All methods provide a quite comprehensive picture on the properties of the composites. STEM analysis identifies the interlayer and EELS gives information on composition. In conclusion the authors point out the importance of the interface for PL properties.

Overall, this is a quite thorough study with many contributors. The PL analysis is performed on a relative scale, quantum yields are not reported.

We would like to thank the referee for their helpful comments. In the revised work, we have included the photoluminescence quantum yield in the main manuscript (Figure 2a), at the same time, we transferred the PL intensity pattern to Supporting Information as Figure S7.

Comments #1: A conceptual question is whether MOF-Perovskite composites are promising for EL or PV applications as the electrically isolating MOF-matrix hampers the charge carrier injection. In this context PL measurements only provide a quite limited information about potential device performance and probably the perspective of such materials is a bit overemphasized.

We appreciate the referee for the constructive comments. We acknowledge that the electron/hole transport behaviour/efficiency of the MOF glass may still lag behind these of the metal oxide used in EL and PV, but there is a growing body of research showing the hybrid materials can be tuned to allow their electron/ionic and hole transport^{21,22}. In addition, the thickness of the MOF glass can be reduced to nano-meter size through advanced fabrication method (like chemical vapor deposition and atomic layer deposition etc). Such advances may allow for practical electron or ion transport. We hope this paper would serve as a guide for future research in device out of this type of composites. Related information was provided in the conclusion part as follows:

“In particular, the notable stability under UV irradiation and relatively high temperature allows these composites to be further processed by an ultrafast laser. The composites generated a localized liquid phase as a result of the intense thermal accumulation, which can be used to fabricate optical products by three-dimensional printing. The whole process can be adjusted by optimizing the pulse duration, repetition rate, and pulse energy as well as ultrafast-laser irradiation time. Furthermore, some advanced fabrication methods like chemical vapor deposition and atomic layer deposition, are promising to reduce the thickness of MOF glass to nano-meter size, making these composites more compatible with planar in device components.”

Comments #2: From a fundamental point of view the study is insightful and worth publication.

We would like to thank the referee for these enthusiastic comments on the novelty, significance, and quality of this work, and are grateful for the constructive comments regarding the perspective of the paper.

References

- 1 Chen, J. *et al.* Efficient and bright white light-emitting diodes based on single-layer heterophase halide perovskites. *Nature Photonics* **15**, 238-244, doi:10.1038/s41566-020-00743-1 (2020).
- 2 Mirijam Zobel, R. B. N., 1Simon A. J. Kimber2*. Universal solvent restructuring induced by colloidal nanoparticles.pdf. *Science*, 292-294 (2015).
- 3 Hou, J. *et al.* Inter marriage of Halide Perovskites and Metal-Organic Framework Crystals. *Angew Chem Int Ed Engl* **59**, 19434-19449, doi:10.1002/anie.202006956 (2020).
- 4 Shin, J. F. *et al.* Self-assembled dynamic perovskite composite cathodes for intermediate temperature solid oxide fuel cells. *Nature Energy* **2**, doi:10.1038/nenergy.2016.214 (2017).
- 5 Luo, J. *et al.* Halide perovskite composites for photocatalysis: A mini review. *EcoMat* **3**, doi:10.1002/eom2.12079 (2021).
- 6 Xu, X. *et al.* High-Performance Perovskite Composite Electrocatalysts Enabled by Controllable Interface Engineering. *Small* **17**, e2101573, doi:10.1002/sml.202101573 (2021).
- 7 Han, T. H. *et al.* Perovskite-polymer composite cross-linker approach for highly-stable and efficient perovskite solar cells. *Nat Commun* **10**, 520, doi:10.1038/s41467-019-08455-z (2019).
- 8 Wang, P. *et al.* Ultrastable Perovskite-Zeolite Composite Enabled by Encapsulation and In Situ Passivation. *Angew Chem Int Ed Engl* **59**, 23100-23106, doi:10.1002/anie.202011203 (2020).
- 9 Sun, K. *et al.* Three-dimensional direct lithography of stable perovskite nanocrystals in glass. *Science* **375**, 307-310 (2022).
- 10 Kim, M. *et al.* Conformal quantum dot-SnO₂ layers as electron transporters for efficient perovskite solar cells. *Science* **375**, 302-306 (2022).
- 11 Jin, H. *et al.* It's a trap! On the nature of localised states and charge trapping in lead halide perovskites. *Materials Horizons* **7**, 397-410, doi:10.1039/c9mh00500e (2020).
- 12 Han, T. H. *et al.* Interface and Defect Engineering for Metal Halide Perovskite Optoelectronic Devices. *Adv Mater* **31**, e1803515, doi:10.1002/adma.201803515 (2019).
- 13 Frohna, K. *et al.* Nanoscale chemical heterogeneity dominates the optoelectronic response of alloyed perovskite solar cells. *Nat Nanotechnol*, doi:10.1038/s41565-021-01019-7 (2021).
- 14 Tan, S. *et al.* Stability-limiting heterointerfaces of perovskite photovoltaics. *Nature* **605**, 268-273, doi:10.1038/s41586-022-04604-5 (2022).

- 15 Zhao, X. *et al.* Accelerated aging of all-inorganic, interface-stabilized perovskite solar cells. *Science* **377**, 307-310 (2022).
- 16 Hu, J. *et al.* Tracking the evolution of materials and interfaces in perovskite solar cells under an electric field. *Communications Materials* **3**, doi:10.1038/s43246-022-00262-2 (2022).
- 17 Huang, Y., Yin, W.-J. & He, Y. Intrinsic Point Defects in Inorganic Cesium Lead Iodide Perovskite CsPbI₃. *The Journal of Physical Chemistry C* **122**, 1345-1350, doi:10.1021/acs.jpcc.7b10045 (2018).
- 18 Nozari, V. *et al.* Ionic liquid facilitated melting of the metal-organic framework ZIF-8. *Nat Commun* **12**, 5703, doi:10.1038/s41467-021-25970-0 (2021).
- 19 Bennett, T. D. *et al.* Melt-Quenched Glasses of Metal-Organic Frameworks. *J Am Chem Soc* **138**, 3484-3492, doi:10.1021/jacs.5b13220 (2016).
- 20 Jingwei Hou, P. C. Liquid-phase sintering of lead halide perovskites and metal-organic framework glasses.pdf. *Science* (2021).
- 21 Wang, W. *et al.* Conductive Polymer-Inorganic Hybrid Materials through Synergistic Mutual Doping of the Constituents. *ACS Appl Mater Interfaces* **9**, 27964-27971, doi:10.1021/acsami.7b09270 (2017).
- 22 Lin Yang, M. P. G., Akanksha K. Menon, Alexandra Bruefach, Kyle Haas, & M. C. Scott, R. S. P., Jeffrey J. Urban. Decoupling electron and phonon transport in single-nanowire hybrid materials for high-performance thermoelectrics. *Science Advance* **7** (2021).

REVIEWERS' COMMENTS

Reviewer #1 (Remarks to the Author):

The authors have addressed our concerns regarding the spectral composition, typo and interfacial fine structures of this series of perovskite-glass hybrid. Combined with their response to other referees, we think now the manuscript is qualified to be accepted by Nature Communications.

Reviewer #2 (Remarks to the Author):

Author has addressed comments well. The manuscript was also updated according to the previously suggestions. I suggest publication of the story.

Reviewer #3 (Remarks to the Author):

The authors have made some substantial changes to address the reviewer's comments, and the quality of the revision has been further improved. I thus have no further comments on the paper and would recommend acceptance of the work.

Reviewer #4 (Remarks to the Author):

The reviewers comments were answered sufficiently.

REVIEWERS' COMMENTS

Reviewer #1 (Remarks to the Author):

The authors have addressed our concerns regarding the spectral composition, typo and interfacial fine structures of this series of perovskite-glass hybrid. Combined with their response to other referees, we think now the manuscript is qualified to be accepted by Nature Communications.

Response:

We are grateful that the reviewer accepted our revised work for publication in Nature Communications.

Reviewer #2 (Remarks to the Author):

Author has addressed comments well. The manuscript was also updated according to the previously suggestions. I suggest publication of the story.

Response:

We express our sincere appreciation for the reviewer's approval of the revised manuscript.

Reviewer #3 (Remarks to the Author):

The authors have made some substantial changes to address the reviewer's comments, and the quality of the revision has been further improved. I thus have no further comments on the paper and would recommend acceptance of the work.

Response:

We appreciate your approval of the revised manuscript for publication in Nature Communications.

Reviewer #4 (Remarks to the Author):

The reviewer's comments were answered sufficiently.

Response:

We express our gratitude for the reviewer's endorsement of our revised manuscript.